# Video-LevelGauge: Investigating Contextual Positional Bias in Video Language Models

**Hou Xia[1], Zheren Fu[1], Fangcan Ling[1], Jiajun Li[2], Yi Tu[2], Zhendong Mao[1]\*, Yongdong Zhang[1]**
[1]University of Science and Technology of China, [2]HUAWEI Technologies Ltd
{overwhelmed,lfc200250}@mail.ustc.edu.cn, {fzr,zdmao,zhyd73}@ustc.edu.cn
{jiajun.work,xssg.tuyi}@huawei.com

## Abstract

Large video language models (LVLMs) have made notable progress in video understanding, spurring the development of corresponding evaluation benchmarks. However, existing benchmarks generally assess overall performance across entire video sequences, overlooking nuanced behaviors such as contextual positional bias, a critical yet under-explored aspect of LVLM performance. We present **Video-LevelGauge**, a dedicated benchmark designed to systematically assess positional bias in LVLMs. We employ standardized probes and customized contextual setups, allowing flexible control over context length, probe position, and contextual types to simulate diverse real-world scenarios. In addition, we introduce a comprehensive analysis method that combines statistical measures with bias pattern recognition to characterize bias. Our benchmark comprises 438 manually curated videos spanning multiple types, yielding 1,177 high-quality multiple-choice questions and 120 open-ended questions, validated for their effectiveness in exposing positional bias. Based on these, we evaluate 27 state-of-the-art LVLMs, including both commercial and open-source models. Our findings reveal significant positional biases in many leading open-source models, typically exhibiting head or neighbor-content preferences. In contrast, commercial models such as Gemini 2.5 Pro show impressive, consistent performance across entire video sequences. Further analyses on context variation, context length, model scale, and multi-modal reasoning provide insights for mitigating bias and guiding model enhancement. Project page: https://github.com/Cola-any/Video-LevelGauge.

## 1 Introduction

Large Video Language Models (LVLMs) have advanced rapidly in recent years, revolutionizing video understanding by integrating large language models (LLMs) with visual perception capabilities. These models (Zhang et al., 2024b; 2025a; OpenAI) show impressive performance across diverse video tasks, with strong generalization abilities. Alongside these advancements, continuous efforts have been devoted to developing video-centric benchmarks to assess their effectiveness, providing critical insights for future improvement.

Existing video benchmarks (Maaz et al., 2024; Liu et al., 2024d) primarily evaluate models based on overall performance across entire video sequences on a variety of tasks, such as temporal reasoning and summarization. Notable benchmarks include MVBench (Li et al., 2024a), TempCompass (Liu et al., 2024d), and MMVU (Zhao et al., 2025) for short videos, as well as MLVU (Zhou et al., 2025) and LongVideoBench (Wu et al., 2024a) for long video understanding. While these benchmarks offer broad task assessments, they provide limited insight into how models interpret content at different contextual positions. As illustrated in Figure 1 (a), LVLMs can suffer from positional bias, i.e., inconsistent comprehension of identical content presented at varying contextual locations. However, existing benchmarks can offer minimal diagnostic value for mitigating this issue.

The serial position effect (Murdock Jr, 1962) in psychology suggests that humans tend to better recall content presented at the beginning and end of a sequence. Similar position sensitivity have

---

*Corresponding Author.

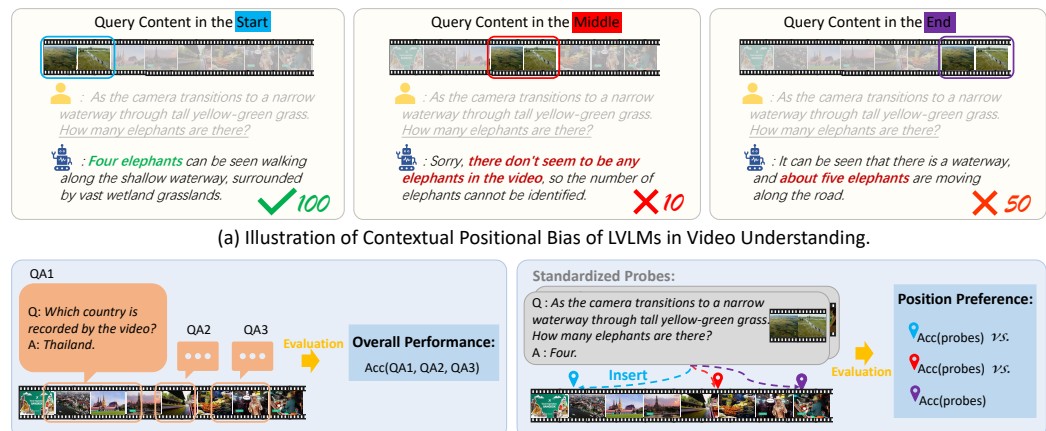

(a) Illustration of Contextual Positional Bias of LVLMs in Video Understanding.

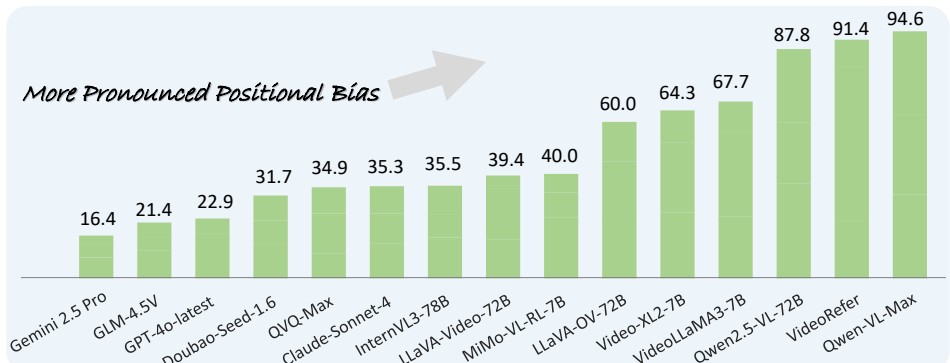

(b) Comparison Between Existing Benchmark (left) and Ours (right).

Figure 1: (a) Large Video Language Models (LVLMs) suffer from positional bias, characterized by uneven comprehension (marked by tick/cross and the score) of identical content presented at different contextual positions. (b) Existing benchmarks typically assess models based on overall performance across the entire video sequence, which is limited in revealing this nuanced behavior. We explicitly investigates this by inserting standardized probes (curated segments tagged with meticulous questions) into varying positions of the context.

Figure 2: **Performance of state-of-the-art LVLMs on Video-LevelGauge**, evaluated by our composite metric, where higher values indicate more pronounced positional bias. Gemini 2.5 Pro (Comanici et al., 2025) exhibits the least positional bias, followed by GLM-4.5V (Team et al., 2025b), GPT-4o-latest (OpenAI), Doubao-Seed-1.6 (Bytedance), and others.

been observed in language models (Liu et al., 2024b) and multi-modal models (Tian et al., 2025; Tan et al., 2024). To date, in video understanding, how various types of LVLMs, such as those incorporating memory components (He et al., 2024) or trained with long-context (Qin et al., 2025), perform on positional biases remains under-explored. Moreover, how positional bias manifests in video–text interleaved contexts is also an open question. In particular, LVLMs claiming to excel at video understanding should be validated for their ability to maintain consistent and effective perception across the entire video sequence, with minimal positional bias.

To this end, we introduce **Video-LevelGauge**, a dedicated benchmark for evaluating contextual positional bias in video understanding. Inspired by needle-in-a-haystack approaches (Nelson et al., 2024; Lu et al., 2025), we propose a standardized probe and customized context strategy for constructing evaluation data. As shown in Figure 1 (b), our benchmark inserts standardized probes at varying positions within the context to assess models' bias to contextual positions. It supports flexible control over context length, probe position, and context composition to evaluate positional biases in various real-world scenarios, such as long video comprehension and multi-modal interleaved inputs. In addition, we propose a comprehensive analysis method for positional bias that combines statistical metrics and bias pattern recognition. Video-LevelGauge includes 438 manually curated

multi-type videos and 1,177 high-quality multiple-choice question answering items spanning six tasks, along with 120 open-ended descriptive questions, constructed through a labor-efficient workflow. These questions are empirically validated to be highly sensitive to visual perception, making them well-suited for detecting positional bias.

We evaluate 27 state-of-the-art LVLMs, including 6 commercial and 21 open-source models based on diverse techniques. We reveal that most leading open-source models suffer from positional bias. They manifest various bias patterns, such as neighbor preference, U-shaped curves, and head preference. In comparison, commercial models, e.g., Gemini 2.5 Pro, achieve consistent and superior performance across the entire sequence, as presented in Figure 2. There are substantial room for improvement in mitigating positional bias among diverse LVLMs, which could significantly enhance their understanding capacity. Our further analysis of context variation, context length, and model scale provides key findings for positional bias in LVLMs. Our contributions are summarized as:

- We emphasize the necessity of positional bias evaluation as a complement to existing benchmarks. To the best of our knowledge, this paper is the first systematic study of contextual positional bias in video understanding.
- We propose Video-LevelGauge, a tailored benchmark for evaluating positional bias in video understanding with a comprehensive analysis method, which supports flexible configurations over context lengths and probe positions.
- We experimentally investigate positional bias in 27 state-of-the-art LVLMs and conduct an in-depth analysis of the effects of context variation, context length, model size and multi-modal reasoning, offering insights for future refinement.

## 2 RELATED WORK

### 2.1 LARGE VIDEO LANGUAGE MODEL (LVLM)

LVLMs have made significant strides with the integration of large language models and visual encoders. Early works (Lin et al., 2024; Maaz et al., 2024) focus on short video understanding through video post-training. Later efforts (Liu et al., 2024a; Li et al., 2024b) extend to long video understanding. Two-stage methods (Zhang et al., 2024a; Park et al., 2025) perform video understanding by inputting frame descriptions into LLMs. Others (Song et al., 2024; He et al., 2024) incorporate explicit memory modules to capture long-term dependencies. Besides, models such as LongVA (Zhang et al., 2025b), LongVILA (Chen et al., 2025) and Video-XL2 (Qin et al., 2025) extend contextual lengths through long video training. More recently, models like Qwen2.5-VL (Bai et al., 2025), InternVL3 (Zhu et al., 2025), and LLaVA-OV (Li et al., 2025) built upon long-context LLMs, exhibit strong performance in long video tasks. MiMo-VL (Team et al., 2025a) and GLM-4.5V (Team et al., 2025b) deliver impressive performance equipped with multi-modal reasoning. Task-specific models have also been developed. VideoRefer (Yuan et al., 2025) enhances fine-grained spatial-temporal understanding and T-Star (Ye et al., 2025) improves contextual search. In this paper, we investigate whether these models can effectively comprehend entire videos without contextual positional bias.

### 2.2 VIDEO BENCHMARK

Alongside the advancement of LVLMs, many efforts have been made to develop video-centric benchmarks. Some benchmarks (Li et al., 2024a; Liu et al., 2024d) focus on evaluating short-video understanding. To extend evaluation to long-videos, VideoMME (Fu et al., 2025), MLVU (Zhou et al., 2025), LongVideoBench (Wu et al., 2024a), LVBench (Wang et al., 2024a), and HLV-1K (Zou et al., 2025) focus on constructing videos of long duration and designing tasks involving long-range comprehension. Another line of work adopts synthetic construction strategies, named Needle-in-a-Haystack (NIAH). For instance, some works (Yuan et al., 2024; Nelson et al., 2024) target long-text comprehension for LLMs, while others (Wang et al., 2025; 2024b; Wu et al., 2024b) focus on multi-modal contexts. More recently, VNBench (Lu et al., 2025), LV-Haystack (Ye et al., 2025), and V-NIAH (Zhang et al., 2025b) are proposed to assess temporal search capabilities in LVLMs. Unlike these benchmarks, which primarily assess models' overall performance across the entire video sequence, our benchmark is specifically designed to evaluate the contextual positional bias in video understanding, providing a dedicated analysis method to reveal nuanced model behaviors.

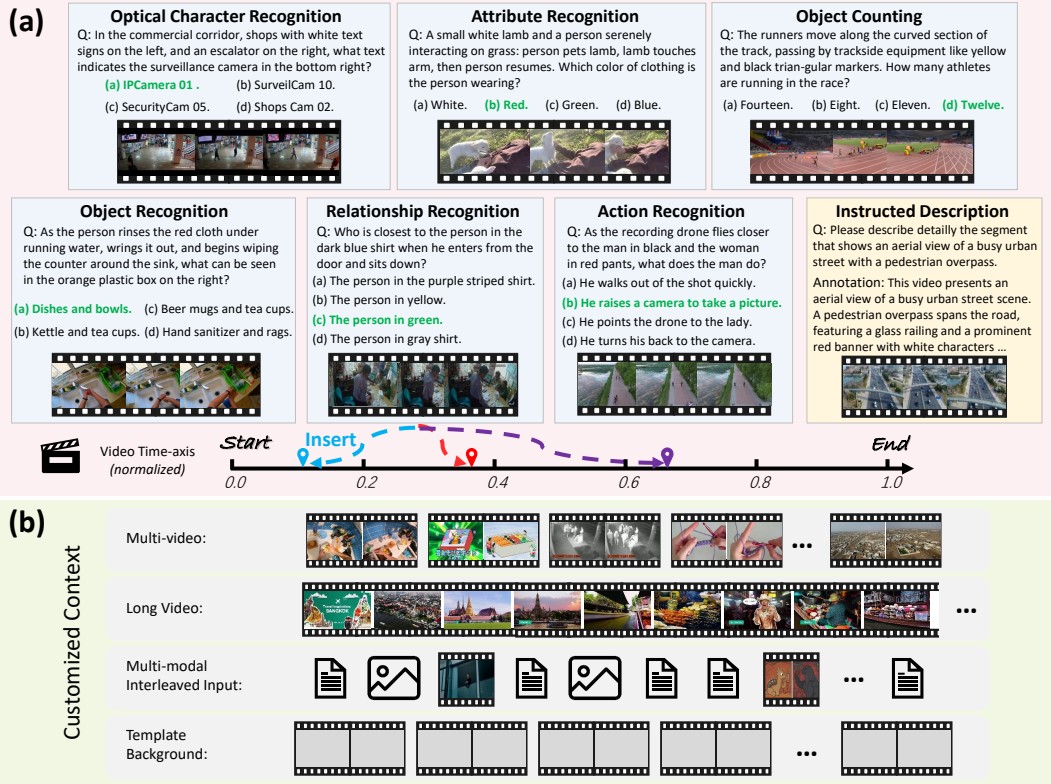

Figure 3: **Overview of Video-LevelGauge**, our benchmark for contextual positional bias in video understanding. It adopts a standardized probe and customized context paradigm, where crafted probes are inserted at varying positions within context. (a) Examples of standardized probes on six multi-choice question answering (MCQA) formatted evaluation tasks and one open-ended instructed description task. (b) Four customized context types for investigating positional bias under various real-world scenarios. *More examples of probes and customized contexts are presented in Sec. A.7.*

## 3 VIDEO-LEVELGAUGE

We begin by presenting our philosophy and statistics (Sec. 3.1), followed by descriptions of probe QA construction (Sec. 3.2), context construction (Sec. 3.3), and our analysis method (Sec. 3.4).

### 3.1 OVERVIEW

As shown in Figure 3, Video-LevelGauge introduces a standardized probe and customized context design paradigm, where carefully designed probe segments are inserted at varying positions within customized contextual contents. By comparing model responses to identical probes at different insertion points, we assess positional bias in video comprehension. Compared to approaches (Zhou et al., 2025; Zou et al., 2025) that densely formulate QA pairs in natural videos, this offers three advantages. (1) Controlled variables, eliminating confounding effects of varying QA difficulty at different positions and contextual information leakage. (2) Flexible control over both the context length and evaluated positions. (3) Simulation of diverse real-world scenarios, such as multi-video un-

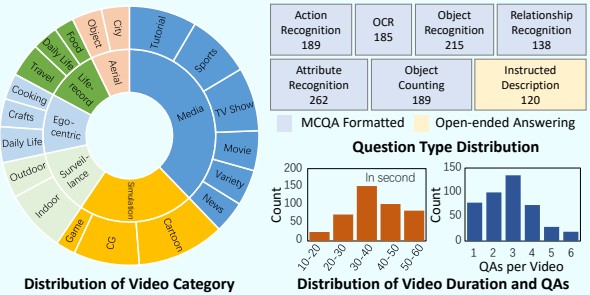

Figure 4: **Benchmark statistics.**

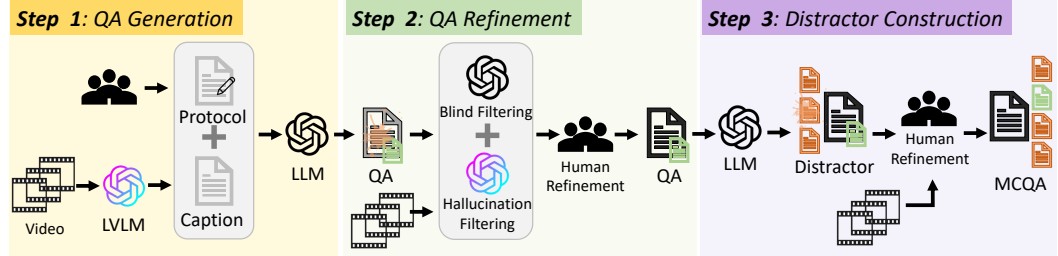

Figure 5: **Probe QA construction workflow.** We propose a labor-efficient workflow that combines automated generation with human refinement. Candidate QAs are first generated using LLM guided by manually defined protocols. These QAs are then enhanced using both LLM and LVLM to eliminate hallucinations and information leakage, followed by human refinement. Finally, for challenge MCQAs, distractor options are generated by LLM and double checked by annotators.

derstanding, long videos, and interleaved video-text inputs. *Beyond synthetic settings, we further construct an extended set by annotating probes on **natural long videos**, detailed in Sec. A.3.*

Video-LevelGauge encompasses six categories of structured video understanding tasks (e.g., action recognition), along with an open-ended descriptive task. As shown in Figure 4, it includes 438 manually collected multi-type videos, 1,177 multiple-choice question answering (MCQA) items, and 120 open-ended instructed descriptive problems paired with annotations. Each question is described with scene mention and task instruction to ensure clarity, requiring genuine visual comprehension. In this way, Video-LevelGauge provides a comprehensive evaluation of positional bias in LVLMs. *We clarify how firmly our compact benchmark supports the evaluation of positional bias in Sec. A.9.*

### 3.2 STANDARDIZED PROBE CONSTRUCTION

The construction of standardized probes involves three stages: video collection, construction of multi-task QA, and probe requirement validation, which are elaborated below.

#### 3.2.1 VIDEO COLLECTION.

As shown in Figure 4, six types of videos are collected from public test sets to avoid data leakage and ethical concerns. Specifically, it includes 42 aerial videos from VisDrone (Zhu et al., 2021), 49 surveillance videos from UCF-Crime (Sultani et al., 2018), 50 egocentric videos from Ego-4D (Grauman et al., 2022), 152 media videos, 58 life-record videos, and 87 synthetic videos from MLVU and VideoMME. PySceneDetect is used to segment the original videos, followed by manual selection. Blurry, static, and duplicate videos are filtered out.

#### 3.2.2 CONSTRUCTION OF MULTI-TASK QA.

To comprehensively evaluate positional bias across various video understanding tasks, we construct six types of structured MCQA tasks: Optical Character Recognition (OCR), Attribute Recognition (AR), Object Recognition (OR), Object Counting (OC), Relationship Recognition (RR), and Action Recognition (ActR), along with one open-ended descriptive task. As shown in Figure 5, we construct probe QAs via a three-step workflow, involving automated generation and human refinement.

*(1) QA Generation*. Collected videos are frame-wise captioned using GPT-4o (OpenAI), and human annotators annotate the task definition along with three positive and negative QA examples for each video task. These video captions, task definitions, and QA examples are then fed into LLMs to generate task-specific question–answer pairs via prompt engineering.

*(2) QA Refinement*. The generated QA pairs may be problematic. We perform blind filtering with LLMs to eliminate questions that leak answers or can be answered using commonsense, and use GPT-4o to filter out items with hallucinations or incorrect answers based on the original videos. Finally, all QA pairs undergo manual refinement.

*(3) Distractor Construction*. For MCQAs, LLMs are prompted to generate multiple deceptive distractor choices, which are then manually vetted and refined by annotators to ensure clarity and no

obvious difference among the options. The choices are randomly shuffled, and the average length of questions is 30.4 words. *Details of human refinement process are provided in Sec A.8.*

### 3.2.3 PROBE REQUIREMENT VALIDATION.

Although blind filtering is applied during QA construction, there may exist multi-modal leakage (Chen et al., 2024). Therefore, we validate the quality of our QAs using Qwen2.5-VL-7B and InternVL3-8B. As shown in Figure 6 left, compared to the needle samples from MLVU, models achieve near-random accuracy (*25%*) under text-only input and perform far from saturation with single-frame inputs in our MC-QAs. This indicates that our QAs are highly sensitive to the degree of visual perception, rendering them well-suited for evaluating positional bias. Furthermore, as shown in Figure 6

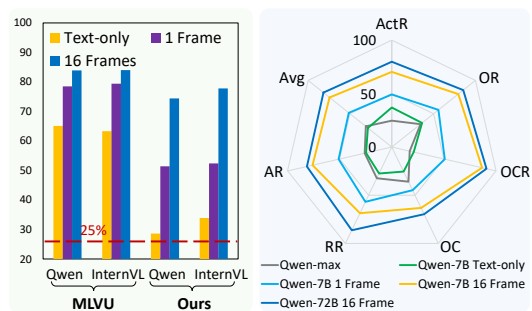

Figure 6: **Probe requirement validation.**

right, the performance of each task scales with both number of frames and model size, highlighting the challenging nature of our QAs to discriminate models of varying capacities. *Validation of potential leakage between the probe clip and background videos is presented in Tab 5.*

## 3.3 CUSTOMIZED CONTEXT CONSTRUCTION

Our design enables customized contextual contents to simulate positional bias under various real-world scenarios: (1) Multi-video understanding. The probe is inserted into a context with multiple videos. (2) Long video understanding. The probe is inserted at specific temporal positions within a natural long-form video. (3) Multi-modal interleaved input. The probe is inserted into an alternating sequence of text and video, a type of context commonly used in retrieval-augmented generation and multi-turn dialogues (Agrawal et al., 2024). (4) Template video background. Inspired by (Xing et al., 2024), a template video initialized with ImageNet mean pixel values is studied as the context.

## 3.4 POSITIONAL BIAS METRIC

Considering the inherent capability variance in models, commonly used absolute accuracy is inadequate for fair cross-model positional bias comparison. To this end, we introduce the relative score (RS) metric by normalizing positional score using model competence, defined as:

$$RS_i = \frac{S_i}{S_{meta}} \tag{1}$$

where $S_i$ is the model accuracy when the probe is inserted at the $i$-th position, and $S_{meta}$ is the accuracy when the probe is provided standalone without any context. Based on this, positional bias is measured by combining both statistical metrics and bias pattern recognition.

For statistical metrics, we propose: Position mean score, $P_{mean} = mean(\{RS_i\}_{i=1}^N)$, where $N$ is the number of evaluated positions, indicates the model's average performance across positions. Position range, $P_{ran} = max(\{RS_i\}_{i=1}^N) - min(\{RS_i\}_{i=1}^N)$, measures the extent of the worst-case variation. Position variance, $P_{var} = Var(\{RS_i\}_{i=1}^N)$, evaluates the positional stability of the model.

To intuitively delineate different bias patterns, we propose Bias Pattern Recognition (*BPR*) to classify models into five types. Stable (—) indicates that model's accuracy maintains consistent across different positions with milder bias. Head preference (↘) indicates that the model's accuracy is higher when probes are inserted at the head of the video, reflecting stronger comprehension of the video's start segment. Neighbor preference (↗) reflects stronger comprehension of the video's tail region, which is closest to the question and model's response in the context sequence. Lost in the middle (U), adopted from prior LLM research Liu et al. (2024b), denotes stronger comprehension of the video's head and tail visual contents and weaker understanding of the middle section. Volatile score (W) represents that the model's performance fluctuates repeatedly with probe insertion positions, showing no regional preference in the video but high sensitivity to positional changes. *Classification is based on polynomial fitting, detailed in Sec. A.5.*

Table 1: **Contextual positional bias analysis of various LVLMs on Video-LevelGauge**. *OCR, AR, OR, OC, RR*, and *ActR* represent six video understanding tasks respectively, as shown in Sec. 3.2. *Average* denotes the average performance of all six tasks. $S_{meta}$: the accuracy with only probe input. $P_{text}$: the text-only score normalized by $S_{meta}$ for convenient comparison with $P_{mean}$; $P_{mean}$: the mean score across all evaluated positions; $P_{ran}$: the range of score between positions; $P_{var}$: the position variance; *BPR*: bias pattern recognition. Metrics are detailed in Sec. 3.4. † represents multi-modal reasoning. Best results are highlighted in color.

| Models | Size | OCR $P_{ran}\downarrow$ | AR $P_{ran}\downarrow$ | OR $P_{ran}\downarrow$ | OC $P_{ran}\downarrow$ | RR $P_{ran}\downarrow$ | ActR $P_{ran}\downarrow$ | $P_{text}$ | $P_{mean}\uparrow$ | Average $P_{ran}\downarrow$ | $P_{var}\downarrow$ | BPR | $S_{meta}\uparrow$ |
|---|---|---|---|---|---|---|---|---|---|---|---|---|---|
| *Two-stages Models* | | | | | | | | | | | | | |
| Caption + Qwen3 | - | 5.6 | 11.7 | 12.9 | 28.3 | 16.9 | 11.3 | 46.5 | 93.1 | 11.2 | 18.1 | U | 65.7 |
| Caption + GPT-4 | - | 4.3 | 11.3 | 7.7 | 12.9 | 15.6 | 7.9 | 45.9 | 96.1 | 7.4 | 5.8 | U | 67.7 |
| *Open-source Models* | | | | | | | | | | | | | |
| MA-LMM | 7B | 19.1 | 16.7 | 20.7 | 9.5 | 9.4 | 6.6 | 78.7 | 83.9 | 9.3 | 7.3 | ↗ | 35.0 |
| LongVA | 7B | 9.6 | 21.6 | 15.8 | 7.7 | 17.1 | 9.1 | 59.1 | 82.4 | 9.2 | 7.8 | U | 48.5 |
| MiniGPT4-Video | 7B | 26.9 | 21.9 | 14.9 | 10.4 | 5.7 | 3.4 | 52.5 | 84.9 | 9.6 | 15.4 | ↘ | 49.3 |
| LLaMA-VID | 13B | 23.2 | 22.0 | 24.2 | 35.5 | 34.5 | 25.4 | 80.5 | 81.1 | 12.3 | 17.3 | W | 31.2 |
| Kangaroo | 8B | 14.3 | 13.2 | 10.4 | 24.6 | 38.9 | 15.6 | 56.0 | 89.2 | 8.2 | 8.4 | W | 53.0 |
| LLaVA-OV | 72B | 3.4 | 7.0 | 5.4 | 4.8 | 15.5 | 6.7 | 43.7 | 92.8 | 5.8 | 3.8 | ↗ | 72.0 |
| LLaVA-Video | 72B | 6.4 | 6.9 | 2.4 | 7.2 | 10.7 | 4.6 | 47.5 | 93.4 | 3.7 | 1.2 | — | 74.6 |
| Qwen2.5-VL | 7B | 8.1 | 27.4 | 9.8 | 16.9 | 31.0 | 15.0 | 41.9 | 89.6 | 12.4 | 11.7 | U | 68.2 |
| Qwen2.5-VL | 72B | 5.6 | 20.2 | 15.8 | 22.2 | 17.7 | 7.3 | 43.6 | 92.2 | 9.1 | 7.0 | ↘ | 73.7 |
| InternVL3 | 8B | 6.7 | 11.5 | 9.9 | 22.6 | 13.9 | 10.7 | 48.1 | 90.3 | 8.0 | 5.3 | U | 70.5 |
| InternVL3 | 78B | 4.7 | 10.6 | 5.0 | 21.4 | 14.7 | 4.5 | 48.5 | 97.1 | 3.9 | 2.8 | — | 74.2 |
| VideoLLaMA2 | 7B | 10.7 | 6.2 | 10.3 | 18.2 | 15.8 | 14.1 | 58.2 | 91.4 | 5.6 | 6.1 | U | 39.3 |
| VideoLLaMA3 | 7B | 4.6 | 7.0 | 6.0 | 9.9 | 13.4 | 9.9 | 48.6 | 89.3 | 5.9 | 3.0 | ↘ | 76.2 |
| NVILA | 8B | 12.5 | 34.2 | 17.2 | 15.1 | 17.7 | 14.0 | 48.9 | 79.1 | 13.9 | 14.7 | ↗ | 64.5 |
| LongVILA-1M | 7B | 7.3 | 22.4 | 14.4 | 10.7 | 12.7 | 12.4 | 44.9 | 81.6 | 11.5 | 12.7 | U | 59.8 |
| Video-XL | 7B | 13.6 | 8.0 | 10.5 | 17.0 | 11.4 | 8.6 | 55.6 | 83.4 | 9.0 | 7.9 | ↘ | 47.7 |
| Video-XL2 | 7B | 3.4 | 13.2 | 6.8 | 13.2 | 7.2 | 6.9 | 48.2 | 91.1 | 6.3 | 3.1 | U | 73.5 |
| VideoRefer | 7B | 4.6 | 9.6 | 7.5 | 8.4 | 11.4 | 7.6 | 40.2 | 80.2 | 5.4 | 2.6 | ↘ | 77.4 |
| T-Star | 7B | 10.9 | 7.1 | 12.6 | 11.9 | 20.5 | 10.6 | 41.2 | 72.6 | 6.1 | 5.0 | W | 69.4 |
| MiMo-VL-RL | 7B | 4.5 | 10.3 | 6.7 | 17.6 | 12.5 | 7.6 | 43.1 | 93.0 | 8.0 | 4.9 | ↘ | 68.9 |
| MiMo-VL-RL† | 7B | 5.4 | 6.4 | 6.5 | 21.5 | 6.9 | 9.0 | 36.9 | 96.8 | 5.8 | 1.8 | — | 69.6 |
| GLM-4.5V | 108B | 4.1 | 3.2 | 2.8 | 8.0 | 6.0 | 8.0 | 33.7 | 97.2 | 3.4 | 1.4 | — | 79.5 |
| GLM-4.5V† | 108B | 3.4 | 5.2 | 1.7 | 6.6 | 4.8 | 6.7 | 30.9 | 97.8 | 2.7 | 1.0 | — | 79.9 |
| *Commercial Models* | | | | | | | | | | | | | |
| Qwen-VL-Max | - | 2.9 | 8.0 | 15.8 | 5.6 | 15.0 | 10.3 | 36.6 | 89.8 | 8.0 | 8.0 | ↘ | 70.5 |
| QVQ-Max† | - | 2.3 | 2.9 | 6.1 | 6.0 | 5.6 | 7.8 | 38.2 | 93.4 | 2.4 | 1.4 | — | 75.8 |
| Doubao-Seed-1.6 | - | 4.9 | 7.0 | 3.5 | 8.9 | 6.5 | 6.2 | 33.2 | 96.9 | 3.2 | 2.4 | U | 80.3 |
| GPT-4o-latest | - | 5.7 | 5.1 | 7.1 | 7.7 | 6.3 | 4.3 | 34.6 | 98.1 | 2.9 | 1.4 | — | 79.9 |
| Claude-Sonnet-4 | - | 7.6 | 9.5 | 8.4 | 5.3 | 7.4 | 7.8 | 37.0 | 96.1 | 3.3 | 2.6 | — | 78.3 |
| Gemini 2.5 Pro† | - | 7.0 | 0.0 | 7.5 | 8.1 | 4.7 | 4.9 | 35.1 | 98.4 | 2.0 | 0.9 | — | 81.7 |

## 4 EXPERIMENTS

We first detail the evaluation protocol, and then present evaluations of a wide range of LVLMs. Next, we analyse the effects of context type, context length, and model size.

### 4.1 EVALUATION PROTOCOL

We conduct a comprehensive investigation of 27 LVLMs using Video-LevelGauge, including 6 commercial models, i.e., Gemini 2.5 Pro (Comanici et al., 2025) and QVQ-Max (Alibaba); 21 open-source LVLMs covering unified models like InternVL3 (Zhu et al., 2025), long video models like Video-XL2 (Qin et al., 2025), specific optimized models like VideoRefer (Yuan et al., 2025), multi-modal reasoning models like GLM-4.5V (Team et al., 2025b), and two-stage methods like LLoVi (Zhang et al., 2024a). Specifically, we evaluate ten uniformly distributed positions across the video context. For each probe, the context is built using nine same type collected videos, resulting in synthetic contexts with an average duration of 7.2 minutes, which are sampled following official implementations of each model. Unless specified, we sample 6 frames per probe and test the model's response by inserting them into different positions, ensuring consistent input of probe frames to the model to isolate sampling impact. *More details of evaluation protocol are provided in Sec. A.6.*

### 4.2 EVALUATION OF SOTA LVLMs

As shown in Table 1, commercial models generally exhibit milder positional bias compared to open-source models. In particular, Gemini 2.5 Pro (Comanici et al., 2025) demonstrates minimal posi-

Figure 7: **Effect of context type on positional bias.** For each model, we visualize the variation in relative scores across probe positions under six types of context (left), and quantify the severity of positional bias using the composite value of three proposed metrics (right), where longer bars indicate more pronounced bias. LVLMs exhibit milder positional bias in simple contexts, such as template videos, and degrade significantly in complex contexts like interleaved video-text inputs.

tional bias (with a small $P_{ran}$ and $P_{var}$, and a stable *BPR*), followed by GPT-4o (OpenAI), Doubao-Seed-1.6 (Bytedance), and QVQ-Max (Alibaba), commercial models known for their long-context capabilities. $S_{meta}$ measures a model's accuracy on the probes and reflects its conventional visual understanding ability, which traditional benchmarks focus on. Although a strong model may be strong in various aspects, $S_{meta}$ is not correlated with positional bias. For example, Qwen-VL-Max (Bai et al., 2023), as an image-oriented model, attains a high $S_{meta}$, yet still exhibits severe positional bias. We speculate that this difference may be attributed to the longer training contexts and larger parameters of commercial models, which can mitigate bias as analysed in Sec. 4.3.

Among the open-source models, two-stage methods that utilize LLMs to interpret sequential frame descriptions exhibit a U-shaped performance, as found by Liu et al. (2024b). The state-of-the-art reasoning models (Team et al., 2025b;a) exhibit minimal positional bias, particularly GLM-4.5V (Team et al., 2025b) with a parameter scale of 10.8 billion. We observe that reasoning patterns can alleviate the positional bias issue to a certain extent (visualized in Figure 14). Besides, Video-LLaMA3 (Zhang et al., 2025a) and LLaVA-Video (Zhang et al., 2024b) are also impressive in terms of reduced positional bias. Video-XL2 (Qin et al., 2025), a model optimized for long videos, exhibits relatively small positional bias. However, LongVA (Zhang et al., 2025b) and LongVILA (Chen et al., 2025) fail to achieve consistently robust performance across the entire sequence. This suggests that the effectiveness of long context training needs to be verified, as analysed in Sec. 4.3.

VideoRefer (Yuan et al., 2025), designed to enhance fine-grained spatial–temporal understanding, shows substantially lower positional bias than models of similar size. Training data with annotation bias is an undeniable contributor. For instance, MiniGPT4-Video (Ataallah et al., 2024), which exhibits head preference bias, is trained on WebVid (Bain et al., 2021), which suffers from static appearance bias (Lei et al., 2023), where visual cues at the beginning of the video are often sufficient for understanding. We also observe that models tend to exhibit lower positional bias on tasks in which they excel. For example, Qwen2.5-VL shows reduced positional bias on the OCR task compared to its bias on other tasks, while MiniGPT4-Video (Ataallah et al., 2024) displays more noticeable bias on OCR and AR tasks. There remains considerable room for improvement in addressing positional bias, and we discuss potential directions in Sec. 4.4. For most models, a clear margin between $P_{mean}$ and $P_{text}$ is observed, indicating that although positional bias exists, the models do not exhibit complete blindness to the context. *Finally, although the positional bias observed in LVLMs exhibits similar patterns in LLMs, mitigating positional bias in LLMs does not appear to directly transfer to LVLMs. A detailed **comparative analysis** is presented in the Sec. A.1.*

### 4.3 FURTHER ANALYSIS

We further investigate the effect of context length, context variation, and model scale, providing actionable insights for future model enhancement.

**Various Customized Contexts.** Our benchmark supports flexible customized contexts. To explore positional bias under different real-world scenarios, we experiment with six distinct context types while keeping the context length fixed. These include: (1) template context, (2) short video context, (3) long video context, (4) multi-video context, (5) long-text context, and (6) interleaved video-text context. These contexts are sourced from public datasets, detailed in Sec. A.4. **Finding 1:**

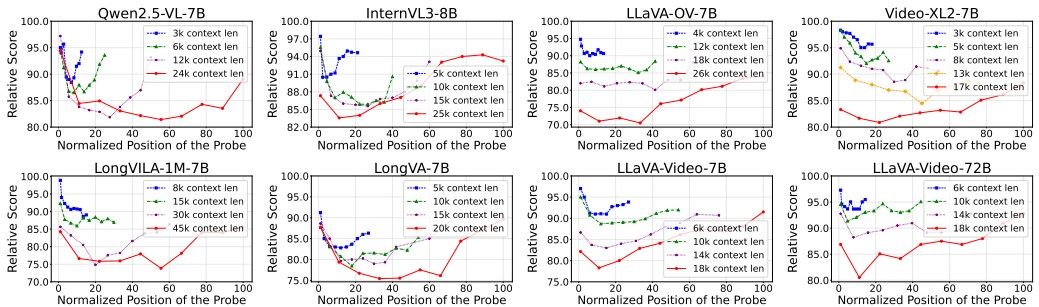

Figure 8: **Effect of context length on positional bias.** Each plot illustrates how positional bias manifests across different context lengths on a specific LVLM. The x-axis denotes position of the probe normalized by the max context length tested, and the y-axis shows the model's relative score. Colored lines represent different context lengths. Positional bias is prevalent across various context lengths. It tends to intensify and the patterns of bias may shift as the context length increases.

**LVLMs exhibit more pronounced positional bias in complex multi-modal context scenarios.** As shown in Figure 7, models generally maintain high performance and exhibit milder positional bias in simpler contexts, such as template and short video context settings. However, as the contextual complexity increases, such as in long video and multi-video scenarios, positional bias becomes more pronounced. Notably, the most severe bias emerges when textual contents are introduced into the context, such as in interleaved video-text and long-text contexts. We attribute this to the lack of training on long mixed-modal context data. This underscores the vulnerability of LVLMs to positional bias when exposed to mixed-modal inputs, which are prevalent in multi-turn dialogue or retrieval-augmented generation. *Refer to Sec. A.2 for more study on* ***hour-long video context.***

**Effect of Context Length.** Models tend to struggle with long-context scenarios, possibly due to inference lengths exceeding the training horizon (Liu et al., 2024c). As shown in Figure 8, we investigate the effect of context length on positional bias across eight representative models, including models built upon long-context LLMs, such as InternVL3 and LongVA, and models specifically optimized via long video training, such as LongVILA-1M (Chen et al., 2025). **Finding 2: Positional bias is prevalent across various context lengths and tends to intensify as the context length increases, accompanied by shifts in bias patterns.** We observe that models such as Video-XL2 (Qin et al., 2025), LongVILA (Chen et al., 2025), and LLaVA-OV (Li et al., 2025) undergo transitions among three patterns of positional bias: a head preference (↘) in shorter contexts, followed by a phase of lost in the middle (U), and eventually a shift toward neighbor bias (↗) in longer contexts. Moreover, models like Qwen2.5-VL (Bai et al., 2025) and LongVA (Zhang et al., 2025b) exhibit U-shaped performance across different lengths. This suggests that LVLMs may inherit the biases of the LLM component. Optimization of positional encoding (Bai et al., 2025) and the effectiveness of long video post-training (Qin et al., 2025) should be verified through positional bias evaluation. Their overall performance gains on videos may stem less from improved comprehension of the entire sequence, and more from improved visual understanding due to exposure to more video data.

**Effect of Model Size.** According to the scaling law (Kaplan et al., 2020), increasing the number of model parameters generally improves performance across a wide range of capabilities. To investigate the effect of model size on positional bias, we conduct experiments on four representative models, as presented in Figure 9. **Finding 3: Positional bias is significantly alleviated as model size increases, consistent with scaling law observed in other capabilities.** Within the same model series, larger variants exhibit more stable performance across the entire sequence, as visualized by flatter bias curves and quantified by lower $P_{var}$ in Table 1. They also show greater resilience to context interference, reflected by consistently higher relative scores. Our experiments

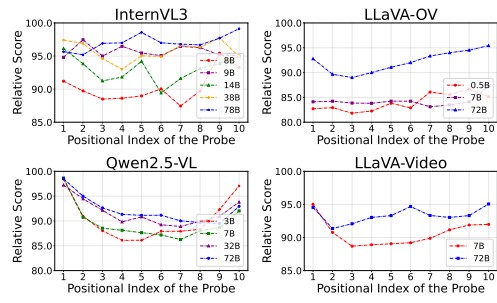

Figure 9: **Effect of model size.**

show that the superior overall performance of larger models on long video tasks stems not only from enhanced comprehension capabilities, but also from their improved handling of blind positions where small ones are more prone to positional bias.

**Open-ended Questions.** Video-LevelGauge also contains 120 open-ended descriptive questions, where LVLMs are instructed to describe a mentioned scene. We use GPT-4 to evaluate model responses by comparing them against annotations. As shown in Table 2, we observe slightly more pronounced positional bias when using descriptive tasks compared to MCQA formatted evaluation. We attribute this to descriptive tasks requiring more fine-grained and comprehensive perceptions, whereas multiple-choice questions typically target a specific object or action, making them relatively easy. As a result, portions of potential positional bias may go undetected in MCQA settings.

Table 2: **Evaluation on open-ended questions.**

| Models | Size | $P_{mean}$ | $P_{ran}$ | $P_{var}$ | $S_{meta}$ |
|---|---|---|---|---|---|
| InternVL3 | 8B | 90.0 | 8.2 | 7.1 | 70.7 |
| LLaVA-Video | 7B | 90.3 | 7.9 | 4.9 | 70.0 |
| LLaVA-OV | 7B | 91.7 | 4.6 | 5.1 | 66.8 |
| Qwen2.5-VL | 7B | 87.5 | 9.4 | 13.8 | 68.3 |
| Qwen2.5-VL | 32B | 92.5 | 9.3 | 7.2 | 72.8 |
| Video-XL2 | 7B | 90.6 | 7.4 | 5.4 | 71.4 |

### 4.4 DISCUSSION.

Contextual positional bias, an inconspicuous but critical issue of LVLMs, may stem from annotation biases in the training data, as prior works (Jung et al., 2025; Yuan et al., 2021) found that models may rely on the linguistic priors of the query for grounding rather than genuine visual understanding. Our analysis suggests that beyond scaling up model parameters, several strategies hold promise for mitigating positional bias, including long video training and enhanced context search algorithms. In addition, we observe amplified positional bias in multi-modal input scenarios, highlighting the potential of training with interleaved video-text data and developing cross-modal context search algorithms. It tends to worsen with longer context lengths; thus, excellent video token compression may not only improve efficiency but also mitigate bias issues. Finally, evaluating positional bias is anticipated to assist in areas of hallucination mitigation and position encoding optimization. The key intuition is that the model should be able to comprehend any part of the context to answer the questions, given that the relevant content may appear anywhere in the sequence.

## 5 CONCLUSION

In this paper, we emphasize the importance of evaluating positional bias in LVLMs. To this end, we introduce Video-LevelGauge, an extensible benchmark tailored for assessing positional bias in video understanding. We systematically evaluate 27 state-of-the-art LVLMs and reveal a disparity in positional bias between commercial and open-source models. Further analysis shows that complex contextual scenarios, especially multi-modal interlaced input, can exacerbate the bias. Moreover, positional bias is pervasive across various context lengths, intensifying and shifting in pattern as the context length increases. Larger model variants tend to be robust. These findings suggest that future model advancement should not only focus on enhancing overall comprehension but also explicitly optimize positional bias, particularly in long video and video-text interleaved understanding tasks.

### ACKNOWLEDGEMENT

This work was supported in part by the Artificial Intelligence-National Science and Technology Major Project under Grant 2023ZD0121200 and in part by the National Natural Science Foundation of China under Grant 62121002. Fundamental and Interdisciplinary Disciplines Breakthrough Plan of the Ministry of Education of China (No. JYB2025XDXM103)

### ETHICS STATEMENT

This work adheres to the ICLR Code of Ethics. In this study, no human subjects or animal experimentation were involved. All datasets used were sourced in compliance with relevant usage guidelines, ensuring no violation of privacy. We have taken care to avoid any biases or discriminatory outcomes in our research process. No personally identifiable information was used, and no

experiments were conducted that could raise privacy or security concerns. We are committed to maintaining transparency and integrity throughout the research process.

## REPRODUCIBILITY STATEMENT

We have made every effort to ensure that the results presented in this paper are reproducible. All code and datasets have been made publicly available to facilitate replication and verification. The experimental setup, including model configurations, and data sources, is described in detail in the paper to assist others in reproducing our experiments.

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

## A   APPENDIX

**Limitations:** Video-LevelGauge is developed for measuring the specific position video understanding capability (i.e., positional bias) of Large Video Language Models. To isolate contextual interference and offer both flexibility and scalability, we adopt a synthetic approach to construct the dataset, facilitating more comprehensive evaluations. This paper actively explores and analyzes the influencing factors of positional bias, as well as the magnitude of bias exhibited by various models under different inputs in real-world scenarios. However, overall, the formation mechanisms and mitigation methods of positional bias remain open questions. On the other hand, traditional video benchmarks undoubtedly hold irreplaceable value. They reflect the model's overall performance on authentic videos. We believe that Video-LevelGauge can serve as a useful complement to existing benchmarks, providing specific and controllable evaluation.

The Appendix is organized as follows:

- Analysis of LVLMs positional bias *vs.* LLMs positional bias (Sec. A.1).
- Validation on hour-long video context (Sec. A.2).
- Extended set constructed on natural long videos (Sec. A.3).
- Details of various customized contexts (Sec. A.4).
- Bias Pattern Recognition (BPR) (Sec. A.5).
- More details of evaluation protocol (Sec. A.6).
- More examples of multi-task QAs (Sec. A.7).
- Detail of human refinement and annotation prompts (Sec. A.8).
- Design criteria of Video-LevelGauge (Sec. A.9)
- Effect of probe length (Sec. A.10).

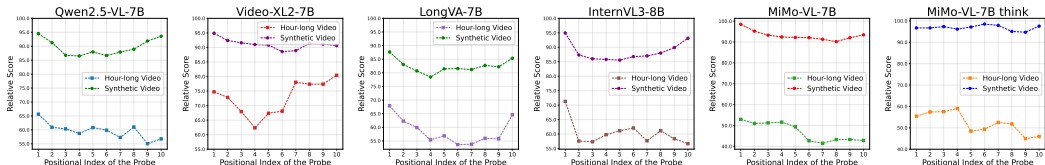

Figure 10: Validation of positional bias on hour-long videos using six LVLMs.

## A.1 LVLMs Positional Bias *vs.* LLMs Positional Bias

As observed, the positional bias in LVLMs exhibits similar patterns in LLMs, such as U-shaped performance. It seems that mitigating positional bias at the LLMs could directly transfer to the LVLMs. However, our experiments indicate that the positional bias in LVLMs differs from that in LLMs. We conduct in-depth analysis on their distinction in light of the experimental results.

- The process of multimodal training of LVLM may substantially alter the positional biases in the LLM component. For instance, as observed in Table 1, Qwen2.5-VL-7B, Video-XL2-7B, VideoRefer, and MiMo-VL-7B are all LVLMs derived from multi-modal training on the Qwen2.5-7B language model, yet they exhibit markedly different patterns of positional bias in video understanding. This suggests that mitigating positional bias in LLMs may not be directly transferable to LVLMs.

- The severity of positional bias in LVLMs is correlated with their visual understanding ability. As analyzed in Sec. 4.2 and illustrated in Figure 13, LVLMs exhibit reduced positional bias on visual tasks in which they excel. This suggests that visual positional bias in LVLMs is, to some extent, attributable to insufficient visual capabilities, which must be acquired through visual learning, similar to the mitigation of the multi-modal hallucination issue.

- The issue of positional bias is more complex in multi-modal scenarios, as interleaved multi-modal inputs can exacerbate positional bias (see Figure 7) and introduce more diverse position/order-related issues (Tan et al., 2024; Tian et al., 2025). In such scenarios, the composition and variability of contextual elements differ substantially: for instance, the embedding discrepancy between two adjacent video tokens is different from that of text tokens. This inherent distinction between textual sequences and multi-modal sequences highlights the need for community attention to multi-modal positional bias.

In summary, similar to other multi-modal capabilities, enhancing the LLM component is expected to improve video positional bias. However, as discussed above, positional bias in multi-modal scenarios deserves special attention, akin to multi-modal positional encoding (Wei et al., 2025).

## A.2 Validation on Hour-long Video Context

We further validate positional bias on hour-long videos. To isolate the influence of frame sampling, we typically fix the frames sampled from the probe, insert them into the context, and then feed the sequence into the model. Here, as a form of real-world validation, we adopt a straightforward setting. We insert the same probe video clip into different positions within a natural long video and construct the entire long video (*.mp4*). The synthetic long videos are then input to the LVLMs, which are sampled by the model's own sampling method. This straightforward setting suffers from two limitations. (1) Although the insertion positions are precise, the content and even the number of frames sampled from the probe by the model may vary for different positions, which can impact study of positional bias; (2) It is inefficient in practice, since a new video file needs to be synthesized and stored temporarily for every tested position. Considering this, we take it as a supplementary validation method rather than as the default setting of our benchmark.

As described in Sec. A.4, 50 long videos are collected from LVBench (Wang et al., 2024a) and HLV-1K (Zou et al., 2025), with an average length of 61.1 minutes. Six representative models are evaluated. Although some models adopt a default fps-based sampling strategy, the video lengths exceed the maximum input limits of the models, leading them to perform uniform frame sampling. Specifically, Qwen2.5-VL, MiMo-VL, Video-XL2, and LongVA sample 768 frames for each hour-long video, InternVL3 samples 128 frames. The results are presented in Figure 10.

It can be observed that, in hour-long video scenarios, LVLMs exhibit more severe positional bias, i.e., lower relative score and greater differences between positions, compared to synthetic video (Figure 8). Among them, Video-XL2 (Qin et al., 2025), which is specifically optimized for long videos, shows relatively higher score on hour-long videos. Comparing the "think" and "no-think" modes of MiMo-VL (Team et al., 2025a), the "think" mode demonstrates a slight improvement in mitigating positional bias. These results are consistent with the observations in the main paper.

### A.3 EXTENDED SET CONSTRUCTED ON NATURAL LONG VIDEOS

Beyond synthetic settings, we further construct an extended set by annotating probes on natural long videos. We first collect 30 long videos from LVBench (Wang et al., 2024a) and HLV-1K (Zou et al., 2025), with an average length of 58.8 minutes. Then, a sliding window of 2 minutes is applied to the long video to extract 10 uniformly distributed, non-overlapping segments as probes. For each segment, annotation is performed following the method described in Sec. 3.2. In total, we obtain 369 MCQAs, with the number of questions being approximately balanced across positions. During evaluation, each video is input to the model ten times, with each run evaluating the model's response to multiple-choice questions regarding one of the ten probes.

Notably, in addition to the lack of flexibility in adjusting probe positions and scaling context length, there are two potential issues in using such data. (1) There are uncontrollable relevant contents of probe segment in natural video context, i.e., information leakage, leading to inaccurate evaluation of positional bias. (2) The quality and number of frames sampled by the model for probe segments may be different, fundamentally impacting the model's understanding of the segment. Therefore, we treat this as an extended set for real-world validation, while our Video-LevelGauge primarily adopts the standardized probe and customized context design paradigm.

We adjust the computation of relative scores in Eq. 1, defined as:

$$RS_i = \frac{S^i_{video}}{S^i_{meta}} \qquad (2)$$

where $S^i_{video}$ is the model accuracy answering MCQAs annotated on the $i$-th position with the entire video as input, and $S^i_{meta}$ is the accuracy when the probe segment is provided standalone without any context. We present $S^i_{meta}$, $i = 1, 2, ..., 10$ in Figure 11. It can be observed that the difficulty of MCQAs constructed on different positions varies, indicating the procedure in Eq. 2 is necessary. Using the extended set, six representative models are evaluated. Specifically, Qwen2.5-VL, MiMo-VL, Video-XL2, and LongVA sample 768 frames for each long video, InternVL3 samples 128 frames. Results are provided in Figure. 12.

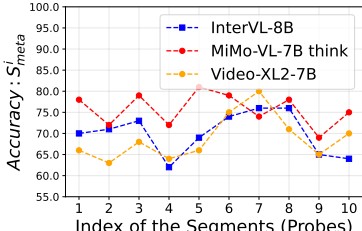

Figure 11: Illustration of $S^i_{meta}$.

Compared with Figure 10, the positional bias observed on natural long videos is obscure. In particular, the differences across positions are smaller, and the bias pattern is not as evident, such as the U-shaped trend. We attribute this to the fact that natural long videos contain many uncontrollable contextual correlations that lead to information leakage, thereby interfering with the measurement of the target position (see LVBench (Wang et al., 2024a) for video cases). To further validate this, we conduct experiments by comparing model response accuracy under three input settings. *Text-only*: only the MCQAs of segments on 3rd position are inputted. *Context*: the visual content of segments on the first position together with the MCQAs annotated on 3rd segments. *Target*: the visual content of 3rd segments together with the corresponding MCQAs. The results are shown in Table 3. *Context* is significantly higher than *Text-only*, indicating the existence of contextual information leakage.

Table 3: Comparison of different input.

| Models | Text-only | Context | Target |
|---|---|---|---|
| InternVL3-8B | 34.7 | 56.5 | 73.9 |
| MiMo-VL-7B | 36.9 | 58.6 | 78.2 |
| Video-XL2-7B | 32.6 | 50.1 | 67.3 |

Figure 12: Evaluation of six LVLMs using the extended set constructed on natural long videos.

## A.4 DETAILS OF VARIOUS CUSTOMIZED CONTEXTS

As shown in Figure 7, we investigate positional bias across six distinct context types while keeping the context length fixed. These include: (1) template context, (2) short video context, (3) long video context, (4) multi-video context, (5) long-text context, and (6) interleaved video-text context. Visual examples of six contexts are illustrated in Figure 15. We detail the configurations as below.

- Template context. We construct the template video by filling all frames with the ImageNet mean RGB pixel values: $[123.675, 116.28, 103.53]$. The template video is uniformly sampled 54 frames as the context. We sample 6 frames per probe and insert them into different positions of context, ensuring the probe frames are consistent to isolate sampling impact.

- Short video context. We randomly sample 100 videos from the short video subset of VideoMME (Fu et al., 2025). During evaluation, each probe is randomly paired with a background video, which remains fixed throughout the positional evaluation. The background video is uniformly sampled 54 frames, into which 6 probe frames are inserted..

- Long video content. We randomly select 50 long videos from long video benchmarks of LVBench (Wang et al., 2024a) and HLV-1K (Zou et al., 2025), using the same processing and evaluation settings as in the short video context. Length statistics of long videos: the shortest is 44.9 minutes, with an average of 61.1 minutes.

- Multi-video context. This setting, described in Sec. 4.1 and Sec. A.6, serves as our default configuration, as it provides flexibility in both context length and complexity. It is an economical and effective approach that enables efficient utilization of video content (probe), while ensuring flexibility and diversity of the context and compactness of the dataset.

- Long-text context. We extract 3,500 paragraphs from OBELICS (Laurençon et al., 2023). To ensure a context length comparable to that of video-based contexts, each paragraph is truncated according to the tested model. For example, for InternVL3, each paragraph is limited to approximately 1,500 tokens to match the token budget of a video clip ($6 \times 256$ tokens). During evaluation, nine paragraphs are randomly selected to form the context, which remains fixed throughout the positional bias assessment.

- Interleaved video-text context. This setting is a variant of the long-text context. During evaluation, each background text is replaced with a video from our benchmark with a probability of 0.5, while all other settings remain consistent.

## A.5 BIAS PATTERN RECOGNITION

To recognize the bias pattern (BPR) of each model, we apply both linear and quadratic fits to the relative score (RS) at all positions. Specifically, we first perform a linear fit to estimate the global trend. Based on the slope and the mean squared error (MSE) of the fit, we clarify BPR into two coarse categories: (1) monotonic type, which includes Stable with milder bias (—), Neighbor preference ($\nearrow$) and Head preference ($\searrow$) with small residuals and a relatively consistent trend; (2) non-monotonic type, which includes Lost in the middle (U) and Volatile score (W), typically exhibiting larger residuals that indicate deviation from linearity. We then apply quadratic fitting to the non-monotonic type to further distinguish structured non-linear patterns.

Given the relative score $\{RS_1, RS_2, \ldots, RS_N\}$, where $N$ is the number of evaluated positions, we fit the data with both linear and quadratic polynomials:

$$\hat{y}^{(1)}(x) = kx + h, \tag{2}$$

$$\hat{y}^{(2)}(x) = ax^2 + bx + c, \tag{3}$$

Figure 13: Illustration of positional bias of Qwen2.5-VL-7B across six video tasks. The x-axis represents the positional index of probes, while the y-axis indicates the model's relative score. Qwen2.5-VL exhibits reduced positional bias on the OCR task compared to its bias on other tasks.

where $\hat{y}^{(1)}(x)$ and $\hat{y}^{(2)}(x)$ denote the fitted functions from the linear and quadratic fits, respectively; $k$ represents the slope coefficient of the linear fit.

Then we calculate the mean squared residuals for both fits separately:

$$\text{MSE}_1 = \frac{1}{n}\sum_{i=1}^{n}\left(RS_i - \hat{y}^{(1)}(x_i)\right)^2, \tag{4}$$

$$\text{MSE}_2 = \frac{1}{n}\sum_{i=1}^{n}\left(RS_i - \hat{y}^{(2)}(x_i)\right)^2. \tag{5}$$

where $\hat{y}^{(1)}(x_i)$ and $\hat{y}^{(2)}(x_i)$ denote the predicted value at the $i$-th position from the linear and quadratic fits, respectively. Based on this, we categorize BPR into five types:

$$\text{BPR} = \begin{cases} — & \text{if } \text{MSE}_1 \leq 3 \text{ and } |k| \leq 0.5, \\ \nearrow & \text{if } \text{MSE}_1 \leq 3 \text{ and } k > 0.5, \\ \searrow & \text{if } \text{MSE}_1 \leq 3 \text{ and } k < -0.5, \\ \text{U} & \text{if } \text{MSE}_1 > 3 \text{ and } \text{MSE}_2 \leq 2, \\ \text{W} & \text{otherwise.} \end{cases}$$

## A.6 EVALUATION PROTOCOL

To isolate the impact of frame sampling and to control the context length (see analysis in Sec. A.2), we sample frames from the probe and the background video separately. Specifically, 6 frames are uniformly sampled from the probe, which remain consistent across different testing positions. The background video is processed using each model's official implementations with 54 frames. The probe frames are then inserted into different positions within the context before being fed into the model. For commercial models, since their sampling strategies are not accessible, we input 6 probe frames and 54 contextual frames jointly in the multi-image format.

We propose three statistical metrics, each capturing a distinct aspect of the model's positional bias. To enable comprehensive evaluation and facilitate comparison across models, we introduce a composite metric (CM), defined as:

$$CM = \alpha_1 P_{ran} + \alpha_2(100.0 - P_{mean}) + \alpha_3 P_{var} \tag{6}$$

where $\alpha_1, \alpha_2, \alpha_3$ are balancing parameters. We empirically set them to 4.0, 3.0, and 4.0, respectively, to ensure that each component contributes comparably to the final score. Higher CM values indicate more pronounced positional bias, as depicted in Figure 2.

## A.7 EXAMPLES OF MULTI-TASK QAS

As shown in Figure 13, model can exhibit varying degrees of positional bias across different tasks. Due to space constraints, only a limited number of multi-task probe examples are included in the main paper. Additional examples of multi-task QAs are provided in Figure 16 and Figure 17. Visual examples of six contexts are illustrated in Figure 15. Following VideoMME, we evaluate multiple-choice QA performance by prompting the LVLM with:

> *Select the best answer to the following multiple-choice question based on the video. Respond with only the letter (A, B, C, or D) of the correct option. QUES-TION, OPTIONS. Answer with the option's letter from the given choices directly.*

For open-ended items, we prompt LVLM with:

> *This video consists of multiple segments. You are tasked to provide a detailed description of the segment described as: QUESTION. Focus only on this part and do not describe other segments. Please describe the visual content of this segment in as detail as possible, including objects, actions, background, and any temporal progression.*

The accuracy of responses for multiple-choice QA is computed locally using exact match. For open-ended questions, following (Maaz et al., 2024), we use GPT-4 as an evaluator. The prompt used for scoring is provided in Figure 24.

### A.8 DETAIL OF HUMAN REFINEMENT AND ANNOTATION PROMPTS

We construct probe QAs through a three-step workflow that combines automated generation with human refinement. Considering video collection process, the construction of Video-LevelGauge involves four major stages requiring human participation.

- *Video Collection.* Diverse videos are manually collected from existing datasets, and blurry, static, and duplicate videos are filtered out. Two annotators participate in this stage, contributing approximately 40 hours of human effort.
- *QA Generation.* Two human annotators cross-validate the annotations, generating the task definition along with three positive and three negative QA examples for each video task.
- *QA Refinement.* Twelve human annotators further review the QAs that pass the model-based blind filtering and hallucination filtering. Annotators are asked to remove or refine QAs that involve answer leakage, commonsense questions, or answers that are inconsistent with the video content. The twelve annotators are divided into two groups. Each group reviews a portion of the data first, and they then cross-check each other's assessments. Over 90 hours of human effort are expended in this stage.
- *Distractor Construction.* Eight annotators examine the LLM-generated multiple-choice questions, filtering out distractors that are trivially distinguishable or are valid given the video content. MCQAs that do not meet quality requirements are LLM-regenerated until they are rejected after three attempts. The eight annotators are also divided into two groups for cross-validation, contributing approximately 60 hours of human effort.

Human annotators are encouraged to focus on validation and refinement rather than rewriting, thereby reducing their workload.

Besides, we elaborate the prompt of automated generation. *(1) QA generation.* The prompts used for frame captioning and QA generation are presented in Figure 18 and Figure 19. *(2) QA refinement.* The prompts used for for blind filtering and hallucination filtering are provided in Figure 20. *(3) Distractor construction.* The prompt employed to instruct LLMs in generating distractors is shown in Figure 21.

Task definitions are provided in Figure 22, and examples of discarded QAs are shown in Figure 23.

### A.9 DESIGN CRITERIA OF VIDEO-LEVELGAUGE

Video-LevelGauge consists of 438 videos and 1,177 MCQAs built upon them. Compared with Video-MME (Fu et al., 2025), which contains 900 videos and 2,700 QAs, its scale is relatively smaller. We clarify that we intentionally keep the number of probes moderate, although constructing more probes does not require substantial annotation labor under the adopted *standardized probe and customized context* paradigm. To evaluate positional bias, each probe needs to be inserted into multiple context positions, thereby expanding the scale of evaluation. For example, with 10 positions by default, evaluating 1,177 QA pairs results in over 10,000 model inferences.

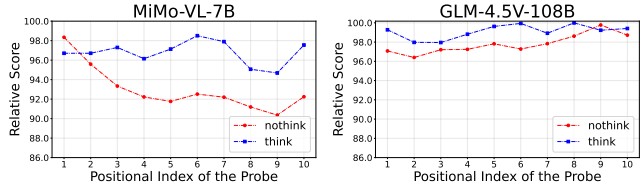

Figure 14: Effect of thinking mode on positional bias. Thinking mode can alleviate the positional bias issue to a certain extent.

Table 4: Effect of probe length. Probe length (PL) is controlled by the number of sampled frames. Although increasing the number of sampled frames significantly improves the absolute performance ($S_{meta}$), the positional bias remains largely unchanged.

| Models | PL | $P_{mean}$ | $P_{ran}$ | $P_{var}$ | $S_{meta}$ |
|---|---|---|---|---|---|
| | 2 | 87.7 | 7.7 | 5.8 | 59.2 |
| InternVL3-8B | 6 | 90.3 | 8.0 | 5.3 | 70.5 |
| | 8 | 90.6 | 8.7 | 5.4 | 74.9 |
| | 2 | 88.1 | 8.0 | 5.6 | 54.3 |
| LLaVA-OV-7B | 6 | 87.3 | 4.2 | 2.1 | 65.6 |
| | 8 | 88.2 | 6.5 | 3.2 | 67.7 |
| | 2 | 90.5 | 11.0 | 11.2 | 60.1 |
| Qwen2.5-VL-7B | 6 | 89.6 | 12.4 | 11.7 | 68.2 |
| | 8 | 90.9 | 10.7 | 10.0 | 72.7 |

This is also why we primarily adopt the MCQA format, as its evaluation is significantly faster than open-end questions. To enhance the usability of the benchmark, we intentionally keep the number of probes moderate, and allocate more effort to improving their quality. For instance, we focus on preventing single-frame or blind-bias predictions, and to suppress potential context leakage (Sec. 3.2.3). These quality-validation efforts are essential for ensuring our reliable assessment of positional bias.

Finally, the probes are highly extensible. They serve as "seeds" that can be flexibly inserted at arbitrary positions, with adjustable context lengths and categories depending on the testing scenario. The key intuition is that the model should be able to comprehend any part of the context to answer the questions, given that the relevant content may appear anywhere in the sequence.

## A.10 EFFECT OF PROBE LENGTH

Furthermore, we examine how probe length affects positional bias by varying the number of sampled frames of each probe (2, 6, and 8). As shown in Table 4, increasing probe length improves the $S_{meta}$, as richer information be sampled. Notably, positional bias basically maintains across different probe length. This suggests that a model's robustness to positional bias is largely independent of probe length and is instead an inherent characteristic of the model.

## B LLM USAGE

Large Language Models (LLMs) were employed to assist in the writing and polishing of this manuscript. Specifically, an LLM was utilized to refine the language, enhance readability, and ensure clarity across various sections of the paper. Its support focused on tasks such as sentence rephrasing, grammar correction, and optimization of the overall textual flow.

Notably, the LLM was not involved in ideation, research methodology design, or experimental planning. All research concepts, core ideas, and analytical work were independently developed and conducted by the authors. The LLM's contributions were limited solely to improving the linguistic quality of the paper, without any involvement in the scientific content or data analysis.

Table 5: Validation of potential leakage between the probe clip and background videos. $S_{meta}$ represents the accuracy when the probe clip is presented to LVLMs independently, without any surrounding context. $B_{1-9}$ represents the accuracy obtained when all nine background videos are presented together with the QA pair, while $B_i$ ($i$=1,..,9) denotes the accuracy when each background video is provided individually along with the QA pair. It is observed that when background videos and the QA pair are provided as input, the model's response accuracy approximates that observed with text-only input. This indicates that the model cannot correctly answer the query question by referring to the background video content, which can be attributed to the manual deduplication performed during video collection (Sec 3.2.1).

| Models | Text-only | $S_{meta}$ | $B_{1-9}$ | $B_1$ | $B_2$ | $B_3$ | $B_4$ | $B_5$ | $B_6$ | $B_7$ | $B_8$ | $B_9$ |
|---|---|---|---|---|---|---|---|---|---|---|---|---|
| Qwen2.5-VL-7B | 28.6 | 68.2 | 32.5 | 28.7 | 31.2 | 30.1 | 29.2 | 31.0 | 30.7 | 30.9 | 29.5 | 30.2 |
| InternVL3-8B | 33.9 | 70.5 | 34.5 | 35.3 | 35.6 | 35.6 | 34.4 | 33.5 | 35.5 | 35.6 | 37.5 | 33.4 |

Table 6: As a supplement to Tab 1, $P_{mean}$ and $P_{ran}$ of each sub-task are reported, providing a clearer view of the overall positional bias for each task.

| Models | Size | OCR | | AR | | OR | | OC | | RR | | ActR | |
|---|---|---|---|---|---|---|---|---|---|---|---|---|---|
| | | $P_{mean} \uparrow$ | $P_{ran} \downarrow$ | $P_{mean} \uparrow$ | $P_{ran} \downarrow$ | $P_{mean} \uparrow$ | $P_{ran} \downarrow$ | $P_{mean} \uparrow$ | $P_{ran} \downarrow$ | $P_{mean} \uparrow$ | $P_{ran} \downarrow$ | $P_{mean} \uparrow$ | $P_{ran} \downarrow$ |
| *Two-stages Models* | | | | | | | | | | | | | |
| Caption + Qwen3 | - | 89.3 | 5.6 | 97.1 | 11.7 | 92.8 | 12.9 | 82.3 | 28.3 | 79.7 | 16.9 | 86.0 | 11.3 |
| Caption + GPT-4 | - | 97.2 | 4.3 | 95.9 | 11.3 | 94.9 | 7.7 | 89.4 | 12.9 | 84.5 | 15.6 | 94.4 | 7.9 |
| *Open-source Models* | | | | | | | | | | | | | |
| MA-LMM | 7B | 87.3 | 19.1 | 90.7 | 16.7 | 76.6 | 20.7 | 90.4 | 9.5 | 83.9 | 9.4 | 82.6 | 6.6 |
| LongVA | 7B | 84.5 | 9.6 | 75.8 | 21.6 | 81.2 | 15.8 | 91.3 | 7.7 | 78.7 | 17.1 | 80.7 | 9.1 |
| MiniGPT4-Video | 7B | 84.6 | 26.9 | 88.6 | 21.9 | 86.3 | 14.9 | 82.8 | 10.4 | 76.6 | 5.7 | 89.4 | 3.4 |
| LLaMA-VID | 13B | 70.9 | 23.2 | 87.5 | 22.0 | 85.2 | 24.2 | 65.2 | 35.5 | 82.7 | 34.5 | 78.8 | 25.4 |
| Kangaroo | 8B | 87.4 | 14.3 | 78.6 | 13.2 | 88.5 | 10.4 | 86.8 | 24.6 | 82.3 | 38.9 | 92.3 | 15.6 |
| LLaVA-OV | 72B | 93.9 | 3.4 | 93.4 | 7.0 | 92.1 | 5.4 | 95.1 | 4.8 | 87.1 | 15.5 | 91.9 | 6.7 |
| LLaVA-Video | 72B | 95.1 | 6.4 | 92.6 | 6.9 | 94.9 | 2.4 | 95.2 | 7.2 | 81.9 | 10.7 | 97.4 | 4.6 |
| Qwen2.5-VL | 7B | 94.8 | 8.1 | 83.9 | 27.4 | 90.0 | 9.8 | 87.9 | 16.9 | 83.9 | 31.0 | 89.1 | 15.0 |
| Qwen2.5-VL | 72B | 95.3 | 5.6 | 89.3 | 20.2 | 94.9 | 15.8 | 90.1 | 22.2 | 87.9 | 17.7 | 95.8 | 7.3 |
| InternVL3 | 8B | 91.7 | 6.7 | 86.8 | 11.5 | 95.0 | 9.9 | 91.4 | 22.6 | 81.9 | 13.9 | 92.9 | 10.7 |
| InternVL3 | 78B | 98.2 | 4.7 | 92.3 | 10.6 | 96.6 | 5.0 | 90.6 | 21.4 | 90.9 | 14.7 | 96.3 | 4.5 |
| VideoLLaMA2 | 7B | 72.4 | 10.7 | 91.1 | 6.2 | 93.4 | 10.3 | 92.5 | 18.2 | 92.1 | 15.8 | 83.4 | 14.1 |
| VideoLLaMA3 | 7B | 91.7 | 4.6 | 88.8 | 7.0 | 95.4 | 6.0 | 92.2 | 9.9 | 75.4 | 13.4 | 92.3 | 9.9 |
| NVILA | 8B | 90.2 | 12.5 | 71.9 | 34.2 | 85.2 | 17.2 | 79.3 | 15.1 | 70.7 | 17.7 | 78.9 | 14.0 |
| LongVILA-1M | 7B | 86.8 | 7.3 | 77.9 | 22.4 | 79.0 | 14.4 | 90.4 | 10.7 | 70.6 | 12.7 | 85.7 | 12.4 |
| Video-XL | 7B | 91.9 | 13.6 | 70.0 | 8.0 | 83.4 | 10.5 | 91.4 | 17.0 | 73.5 | 11.4 | 87.3 | 8.6 |
| Video-XL2 | 7B | 97.9 | 3.4 | 84.4 | 13.2 | 91.6 | 6.8 | 90.6 | 13.2 | 85.6 | 7.2 | 95.9 | 6.9 |
| VideoRefer | 7B | 85.1 | 4.6 | 78.8 | 9.6 | 83.7 | 7.5 | 76.7 | 8.4 | 71.1 | 11.4 | 87.6 | 7.6 |
| T-Star | 7B | 62.3 | 10.9 | 68.8 | 7.1 | 68.2 | 12.6 | 61.5 | 11.9 | 71.5 | 20.5 | 74.1 | 10.6 |
| MiMo-VL-RL | 7B | 94.5 | 4.5 | 89.6 | 10.3 | 94.0 | 6.7 | 88.0 | 17.6 | 90.5 | 12.5 | 94.5 | 7.6 |
| MiMo-VL-RL† | 7B | 97.2 | 5.4 | 96.0 | 6.4 | 98.7 | 6.5 | 89.8 | 21.5 | 97.9 | 6.9 | 94.7 | 9.0 |
| GLM-4.5V | 108B | 96.8 | 4.1 | 98.8 | 3.2 | 99.0 | 2.8 | 95.0 | 8.0 | 97.3 | 6.0 | 97.4 | 8.0 |
| GLM-4.5V† | 108B | 98.2 | 3.4 | 94.4 | 5.2 | 99.4 | 1.7 | 98.4 | 6.6 | 97.9 | 4.8 | 94.2 | 6.7 |
| *Commercial Models* | | | | | | | | | | | | | |
| Qwen-VL-Max | - | 90.2 | 2.9 | 82.3 | 8.0 | 93.2 | 15.8 | 95.2 | 5.6 | 84.0 | 15.0 | 95.4 | 10.3 |
| QVQ-Max† | - | 94.2 | 2.3 | 95.3 | 2.9 | 88.9 | 6.1 | 90.1 | 6.0 | 90.0 | 5.6 | 93.7 | 7.8 |
| Doubao-Seed-1.6 | - | 95.1 | 4.9 | 94.3 | 7.0 | 98.2 | 3.5 | 95.2 | 8.9 | 94.0 | 6.5 | 98.4 | 6.2 |
| GPT-4o-latest | - | 96.6 | 5.7 | 97.2 | 5.1 | 99.0 | 7.1 | 98.1 | 7.7 | 94.4 | 6.3 | 98.4 | 4.3 |
| Claude-Sonnet-4 | - | 95.9 | 7.6 | 94.3 | 9.5 | 96.6 | 8.4 | 98.8 | 5.3 | 97.0 | 7.4 | 93.9 | 7.8 |
| Gemini 2.5 Pro† | - | 99.4 | 7.0 | 99.4 | 0.0 | 98.3 | 7.5 | 97.4 | 8.1 | 98.0 | 4.7 | 97.9 | 4.9 |

The authors take full responsibility for the content of manuscript, including all text generated or polished with the LLM's assistance. We confirm that all text produced with the LLM's support complies with ethical guidelines and does not lead to plagiarism or any form of scientific misconduct.

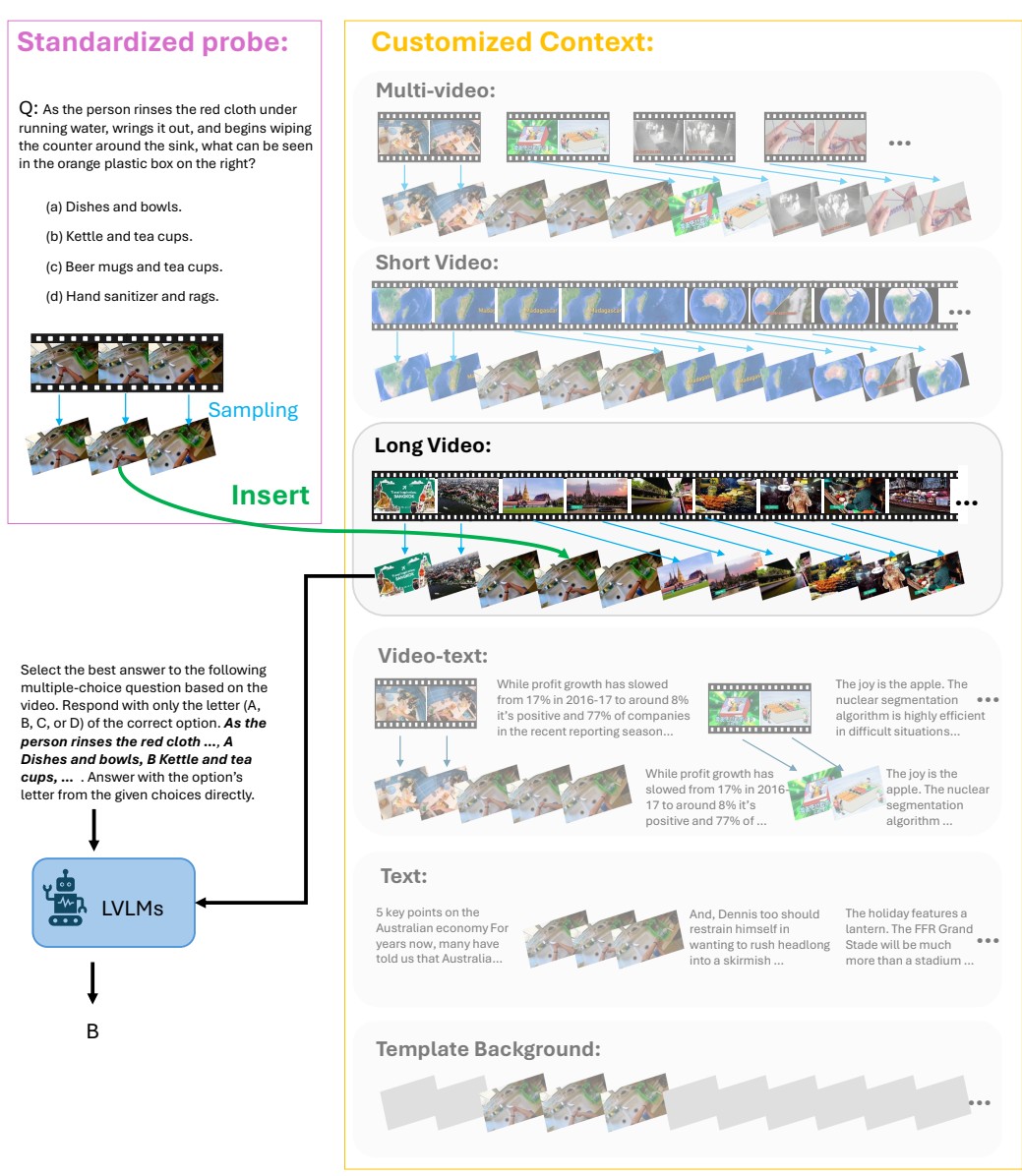

Figure 15: Visual examples of customized contexts. We illustrate six types of customized contexts and how the probe is inserted into each of them. Refer to Sec A.4 for more details.

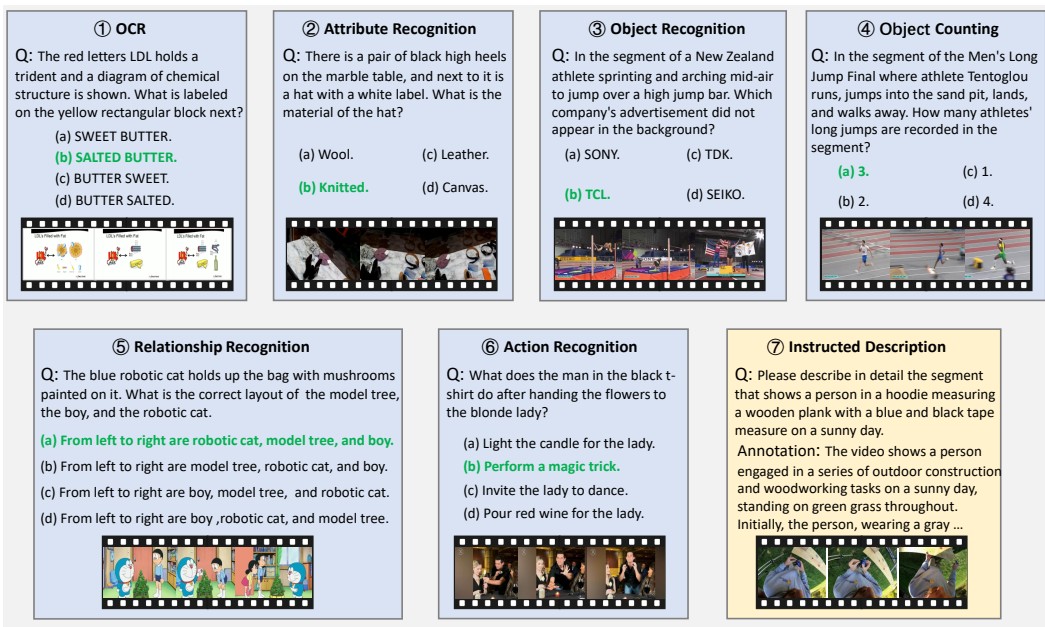

Figure 16: Examples of probes for six MCQA formatted evaluation tasks and one open-ended instructed description task. Each question is described with scene mention and task instruction to ensure clarity, requiring genuine visual comprehension.

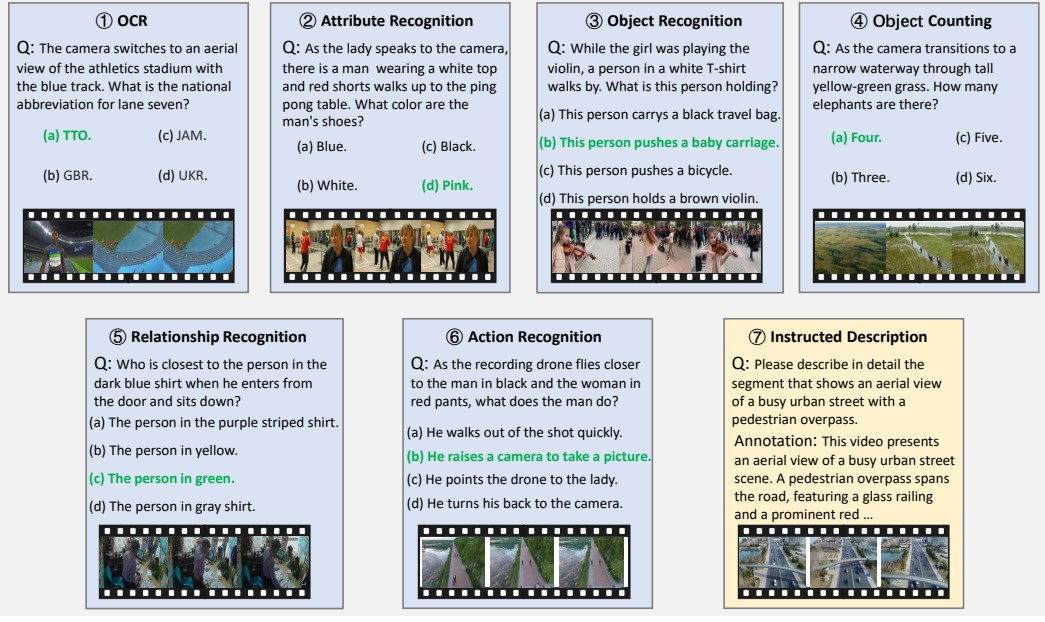

Figure 17: Examples of probes for six MCQA formatted evaluation tasks and one open-ended instructed description task. Each question is described with scene mention and task instruction to ensure clarity, requiring genuine visual comprehension.

**Frame Caption Prompt**

You are a meticulous and detail-oriented video frame analyzer. Your core task is to provide an exhaustive, objective, and factually accurate description of the given frame, strictly based on observable content without any subjective inferences or assumptions.

Please adhere to the following guidelines when describing the frame:
**1. Detailed Object Description**
    Identify all distinct objects present (e.g., people, furniture, natural elements, text, etc.).
      For each object, specify:
- **Quantification:** Exact or approximate number (e.g., "three chairs," "a single tree").
- **Physical properties:** Color (e.g., "maroon with white stripes"), shape (e.g., "circular," "rectangular"), size relative to other objects (e.g., "smaller than the table"), texture (e.g., "smooth metal," "fuzzy fabric"), and any notable features (e.g., "a cracked vase," "a sign with faded letters").
- **State:** If applicable, describe the object's condition (e.g., "upright," "broken," "moving").

**2. Relationships Between Objects**
- **Spatial relationships:** Clearly define positions relative to one another (e.g., "the lamp is on top of the desk," "the person is standing to the left of the door," "the cup is beside the book"). Include distance cues where observable (e.g., "closely adjacent," "several feet apart").
- **Interaction or connection:** Note if objects are in contact, attached, or interacting (e.g., "the hand is holding the pen," "the wire is plugged into the socket," "the rain is falling on the roof").

**3. Language and Objectivity**
- Use clear, fluent, and precise natural language. Avoid ambiguous terms (e.g., "nearby" can be replaced with "immediately next to" if applicable).
- Strictly base descriptions on what is visually verifiable. Do not include guesses about intent, context, or unobserved details (e.g., instead of "the person looks sad," describe "the person's mouth is downturned and eyebrows are furrowed").

Your goal is to create a description that is so thorough a reader could visualize the frame accurately without seeing it.

Figure 18: The collected videos are captioned frame-wise at 1 FPS using GPT-4o with this prompt. We instruct the model to generate detailed descriptions of each frame, encompassing elements such as the number of objects, visible text, and object relationships. This detailed captioning serves as a foundation for subsequently generating multi-task question-answer pairs.

---

**QA Generation Prompt**

You are tasked with creating 2-3 highly challenging questions that test memory recall of specific details from a video. The descriptions of the video is provided by frame-wise. You need to create questions defined as follow.
Question Type definition: <TASK_NAME> : <TASK_DEFINITION>.

The video is described in frames as follows:
Frame 1: <FRAME1>
Frame 2: <FRAME1>
…

The generated questions need to STRICTLY meet the following requirements:
- The type of question meets the Question Type definition: <TASK_NAME> : <TASK_DEFINITION>.
- The question is about the video content.
- Questions should BRIEFLY refer to unique scenes, events, or characters, avoiding vague descriptions.
- The question should be as challenging as possible and can only be answered by relying on the video content, rather than commonsense.

Here are two POSITIVE example questions:
Positive Example1: <POSITIVE_QA1> ;
Positive Example2: <POSITIVE_QA2> ;
Positive Example3: <POSITIVE_QA3>.

Here are two NEGATIVE example questions:
Negative Example1: <NEGATIVE_QA1> ;
Negative Example2: <NEGATIVE_QA2> ;
Negative Example3: <NEGATIVE_QA3>.

Please output the questions and reference answers in the following JSON format: [
    {'question': 'xxx', 'answer': 'xxx'},
    {'question': 'xxx', 'answer': 'xxx'},
    …
]

Figure 19: Based on detailed video captions and human-annotated protocols, GPT-4 is instructed, using this prompt, to generate 2–3 task-specific question-answer pairs for each video. Different types of questions are generated separately. In total, we obtain 7,319 candidate QAs at this stage. Definitions of each task are presented in Figure 22.

**Blind Answer Prompt**

Your task is to determine whether the given Question meets the provided Question Category Definition and then briefly answer the Question.

First, please carefully read the following definition of the question category:
Question Category definition: <TASK_NAME> : <TASK_DEFINITION>.

Next, please read the following Question: <QUESTION>.

Now, follow these steps:
- Analyze whether the example question meets the category definition and state your conclusion (Yes/No).
- Use your knowledge to briefly answer the example question, or guess an answer even if your knowledge cannot answer it.

Please output your answers in the following JSON format: {'MeetsCategory': 'yes/no', 'answer': 'xxxxx'}"

**Filtering Prompt**

Your task is to evaluate whether the given answer to be evaluated is consistent with the reference answer and correctly answers the question.

First, please carefully read the following question: <QUESTION>.

Next, read the reference answer: <REFERENCE_ANSWER>.

Now, read the answer to be evaluated: <LLM_ANSWER>.

When evaluating the answer to be evaluated, please follow these steps:
- Carefully understand the question.
- Compare the reference answer with the answer to be evaluated.
- Check if the answer to be evaluated is consistent with the reference answer.

You should answer with Yes/No first, and Provide a brief explanation here .

**LVLM Answer Prompt**

Your task is to evaluate whether the provided answer correctly responds to the question based exclusively on the given video content.

Video content: <VIDEO_CONTENT>.

Here is the question: <QUESTION>.

And here is the answer to be evaluated: <REFERENCE_ANSWER>.

You should carefully examine the video content. Fully understand the question and answer and make judgment.
Judgment Criteria:
- Correct: Answer is fully consistent with video content and completely addresses the question
- Incorrect: Answer contradicts video content, adds unsupported information, or fails to address the question.

You should answer with Correct/Incorrect first, and Provide a brief explanation here.

Figure 20: During QA refinement, the generated questions are submitted to GPT-4 for blind answering using the Blind Answer Prompt. Subsequently, the QA pairs and blind answers are evaluated with the Filtering Prompt, which filters out questions answerable using text-only input. Additionally, GPT-4o is tasked with verifying the correctness of the QAs based on the original video content. Following manual selection, we obtain 1,177 final QAs and 120 descriptive items. While the construction of open-ended items has been completed, distractor generation for multiple-choice QAs requires an additional step.

**Distractor Prompt**

Your task is to generate three different seemingly reasonable interference answers based on the form of the given question and the correct answer.

First, please carefully read the following question: <QUESTION>.

And here is the correct answer: <REFERENCE_ANSWER>.

When generating the interference answers, please note the following guidelines:
- The interference answers should be in a similar form or context as the correct answer.
- They should seem reasonable at first glance, but not be the correct solution to the question.
- Avoid using completely off - topic or obviously wrong answers.

Please output your response in the following JSON format with English: [
    'interference answer1', 'interference answer2', 'interference answer3'
]

Figure 21: For MCQAs, we input the verified question-answer pairs into GPT-4 and use a specific prompt to generate three distractors. This step is critical for creating high-quality, challenging multiple-choice questions, as low-quality distractors may lead to information leakage, for instance, enabling model to guess the correct answer easily by elimination.

**Task Definition**

"**Object Recognition** ": "Briefly describe the scene and specify the location of an object, and then ask a question about the object(s) is(are).",

"**Attribute Recognition** ": "A question that briefly describe the scene and specify an object, and then ask to identify a property of the object such as color, shape, material, etc.",

"**Object Counting**": "A question that briefly describe the scene and specify a certain type of object/action, and then ask to identify the number of objects/actions",

"**Relationship Recognition**": "A question that briefly describe the scene and specify several objects, and then ask to identify the relationship between them.",

"**OCR**": "A question that briefly describe the scene and specify an area, and then ask to identify the text content in that area. It can be subtitles or any other text that appears in the video",

"**Action Recognition** ": "A question that briefly describe the scene and specify an object(human) and then ask about the action of the object(human).",

Figure 22: Task definitions for six video understanding tasks. Additionally, descriptive questions are formulated as instructions that prompt the model to generate detailed descriptions of the video segments specified by the scene mentions.

Figure 23: Three representative examples of filtered question–answer pairs are presented. The first example illustrates a question answerable through commonsense knowledge. The second exemplifies answer leakage originating from the scene description. The third demonstrates a hallucinated response, potentially caused by inaccuracies in frame descriptions generated in Step 1.

Figure 24: We use GPT-4 to evaluate model generated open-ended outputs by comparing them against annotations.

