# OpenReview forum: "Video-LevelGauge: Investigating Contextual Positional Bias in Video Language Models."
_ICLR.cc/2026/Conference — ICLR 2026 Poster_

### Official Review · Reviewer_sdzE · 2025-10-23

**Soundness:** 2
**Presentation:** 3
**Contribution:** 3
**Rating:** 6
**Confidence:** 3

**Summary:**

The authors address the problem of **contextual positional biases** in large video-language models (LVLMs), where the content of a clip is interpreted inconsistently depending on its position within a video. While this issue has been explored in language models, it has not been studied in multimodal video-language settings, making this work both unique and novel. The paper introduces **Video-LevelGauge**, a benchmark designed to evaluate contextual positional biases in LVLMs across diverse tasks and video types. Beyond standard accuracy, it proposes novel **statistical metrics** to quantify such biases. Using these metrics, the authors further characterize **morphological patterns** in LVLMs, enabling the identification of specific phenomena underlying positional bias. Extensive experiments and analysis conducted with Video-LevelGauge provide valuable insights to guide future LVLM development and contextual positional bias mitigation. Overall, the paper is well-written and presents notable contributions to a previously underexplored area of LVLM research.

**Strengths:**

- Video-LevelGauge incorporates a comprehensive coverage of six video-language tasks spanning diverse video types, including egocentric, media, and synthetic videos, enabling extensive evaluation of existing LVLMs for contextual positional biases across a multitude of settings.
- Video-LevelGauge introduces three statistical metrics: $P_{mean}$, which captures the magnitude of positional bias, and $P_{ran}$ and $P_{var}$, which measure the volatility of model behavior. Together, these metrics provide a holistic assessment of positional biases in LVLMs.
- The proposed morphological recognition provides a grounded approach to identifying potential root causes of contextual positional biases, providing a framework to diagnose and characterize positional biases in LVLMs.
- Thorough analysis is performed on Video-LevelGauge using the proposed metrics across multiple factors, including context length and video type, revealing several insightful observations that could inform future efforts toward mitigating these biases.

**Weaknesses:**

- For the positional bias metric, the relative score is intuitive and easy to understand. However, greater details could be provided on what the score entails: How is $RS_i$ computed if $S_{meta} = 0$, and what does $RS_i$ reflect in this case? Additionally, for the five types of morphological patterns, several categories are not straightforward to understand. In particular, the morphological phenomena of “lost in the middle”, “neighbor bias”, and “volatile” could be somewhat confusing. I would suggest for the authors to perhaps provide additional elaboration for each of these types in Section 3.4, and possibly how they correlate with model performance.
- Although the authors validated against multimodal information leakage within each probe instance, the paper does not seem to address potential leakage between the probe clip and background video(s) during evaluation. Background videos with similar contexts to the probe clip could lead to cross-video leakage, allowing the LVLM to answer the query correctly by referring to background video content, which may be unintended.
- Evaluation results for individual tasks (e.g., OCR, AP, etc.) only report $P_{ran}$, which measures the worst-case variation, but omit $P_{mean}$, which captures the average performance across instances. Including $P_{mean}$ would provide a clearer view of the overall positional bias for each task.

**Questions:**

- “$\nearrow \text{if MSE}_1 \leq 3 \text{ and }k > 0.5$” [Appendix A.5] - Could the authors elaborate what the neighbor bias entails, and how the trend reflected by this condition correlates with this bias?

---

> ### Author Response · Authors · 2025-11-25
> **Response to Reviewer sdzE (1/2)**
>
> Dear reviewer sdzE, thank you for your insightful comments. We address your comments as follows and have incorporated the corresponding revisions into the manuscript (highlighted in blue).
>
> ---
> **Q1:** For the positional bias metric, the relative score is intuitive and easy to understand. However, greater details could be provided on what the score entails: How is $RS_{i}$ computed if $S_{meta}=0$, and what does $RS_{i}$ reflect in this case? Additionally, for the five types of morphological patterns, several categories are not straightforward to understand. In particular, the morphological phenomena of “lost in the middle”, “neighbor bias”, and “volatile” could be somewhat confusing. I would suggest for the authors to perhaps provide additional elaboration for each of these types in Section 3.4, and possibly how they correlate with model performance.
>
> **A1:** $S_{meta}$ represents the accuracy when the probe clip is presented to LVLMs independently, without any surrounding context. We would like to clarify that $S_{meta}$ would not be zero in the practical evaluation. On one hand, since our benchmark consists of four-choice question answering, even random guessing can theoretically yield an accuracy of approximately 25%. On the other hand, an extremely low $S_{meta}$ indicates that the model lacks basic visual understanding, which renders the evaluation of positional bias uninformative.
>
> We appreciate the reviewer’s comment highlighting this potential source of confusion. We clarify the morphological phenomena as follows and have accordingly improved the description in Sec. 3.4.
>
> - "**Lost in the middle**" indicates that the model’s accuracy is higher when probes are inserted at the beginning or end of the video context compared to when probes are positioned in the middle. This reflects stronger comprehension of the video’s head and tail visual contents and weaker understanding of the middle section. The term “lost in the middle” is adopted from prior LLM research [1].
>
> - "**Neighbor preference**" indicates that the model’s accuracy is higher when probes are inserted at the end of the video, reflecting stronger comprehension of the video’s tail region. We refer to this as “neighbor preference” because the end of the video content is proximal to the query question and the model’s response in the context sequence.
>
> - "**Volatile**" indicates that the model’s performance fluctuates repeatedly with probe insertion positions, exhibiting no clear regional preference within the context while showing high sensitivity to positional changes.
>
> [1] Nelson F. Liu, Kevin Lin, John Hewitt, Ashwin Paranjape, Michele Bevilacqua, Fabio Petroni, and Percy Liang. Lost in the middle: How language models use long contexts. Transactions of the Association for Computational Linguistics, 12:157–173, 2024b.

---

> ### Author Response · Authors · 2025-11-25
> **Response to Reviewer sdzE (2/2)**
>
> **Q2:** Although the authors validated against multimodal information leakage within each probe instance, the paper does not seem to address potential leakage between the probe clip and background video(s) during evaluation. Background videos with similar contexts to the probe clip could lead to cross-video leakage, allowing the LVLM to answer the query correctly by referring to background video content, which may be unintended.
>
> **A2:** Thank you for this insightful comment. Contextual information leakage is a critical concern in our benchmark. As analyzed in Sec. A.3, constructing QAs on natural videos can inevitably suffer from contextual information leakage. We adopt the *standardized probe and customized context* design paradigm to control such leakage effectively. During video collection (Sec. 3.2.1), manual deduplication is performed, and background videos are selected from these curated videos, thereby effectively reducing the homogeneity of contextual information.
>
> As suggested, we evaluate the accuracy of the model when the background video(s) and QA pair are provided as input. Results are shown below (added to the revision as Table 5):
> | Models           | Text-only | $S_{meta}$ | $B_{1-9}$ | $B_{1}$  | $B_{2}$  | $B_{3}$   | $B_{4}$   | $B_{5}$   | $B_{6}$   | $B_{7}$   | $B_{8}$   | $B_{9}$   |
> |-----------------|-----------|---------|------|------|------|------|------|------|------|------|------|------|
> | Qwen2.5-VL-7B   | 28.6      | 68.2    | 32.5 | 28.7 | 31.2 | 30.1 | 29.2 | 31.0 | 30.7 | 30.9 | 29.5 | 30.2 |
> | InternVL3-8B    | 33.9      | 70.5    | 34.5 | 35.3 | 35.6 | 35.6 | 34.4 | 33.5 | 35.5 | 35.6 | 37.5 | 33.4 |
>
> **Table:** $S_{meta}$ denotes the accuracy when the probe clip and QA are input to the LVLMs. $B_{1-9}$ represents the accuracy obtained when all nine background videos are presented together with the QA pair, while $B_{i}$  ($i$=1,..,9) denotes the accuracy when each background video is provided individually along with the QA pair.
>
> It is observed that when background videos and the QA pair are provided as input, the model’s response accuracy approximates that observed with text-only input. This indicates that the model cannot correctly answer the query question by referring to the background video content.
>
> ---
> **Q3:** Evaluation results for individual tasks (e.g., OCR, AP, etc.) only report $P_{ran}$, which measures the worst-case variation, but omit $P_{mean}$, which captures the average performance across instances. Including $P_{mean}$ would provide a clearer view of the overall positional bias for each task.
>
> **A3:** In the initial draft, we reported only $P_{ran}$ for each sub-task to avoid making Table 1 overly crowded. As suggested, we have included the $P_{mean}$ values for all six sub-tasks in Table 6 in the revised manuscript.
>
> ---
> **Q4:**  "$\nearrow  \text{if } \mathrm{MSE}_1\leq3 \text{ and } k>0.5, $" [Appendix A.5] - Could the authors elaborate what the neighbor bias entails, and how the trend reflected by this condition correlates with this bias?
>
> **A4:** "Neighbor preference" indicates that the model’s accuracy is higher when probes are inserted at the end of the video, reflecting stronger comprehension of the video’s tail region. In our analysis, we apply both linear and quadratic fits to recognize the morphological pattern (MR) of each model. If the variance of the linear fit meets $MSE_{i} ≤ 3$, indicating that the local changes are relatively stable, we classify the pattern as linear-like, which includes milder bias (—), neighbor preference ($\nearrow$), and head preference ($\searrow$). We then examine the slope $k$ to determine the specific direction of the trend: if $k > 0.5$, we identify it as a neighbor preference ($\nearrow$) pattern.
>
> ---
> We deeply appreciate the reviewer's time and thoughtful suggestions. Should the reviewer have any remaining questions or require further details, we would like to provide them.

---

> > ### Comment · Reviewer_sdzE · 2025-11-27
> > **Response to authors**
> >
> > I don't have major concerns after reading the authors' rebuttal.
> >
> > In the meantime, I also read some weaknesses from other reviewers' concerns. I will keep my borderline accept score, yet may not champion during the discussion stage with other reviewers.

---

### Official Review · Reviewer_TpSR · 2025-10-27

**Soundness:** 2
**Presentation:** 1
**Contribution:** 3
**Rating:** 2
**Confidence:** 4

**Summary:**

This paper introduces a benchmark suite for evaluating contextual positional bias of large video language models (LVLMs) considering long contexts. The data collection involves three stages,
1. QA Generation: They collect videos from existing video-text resources including ego-centric videos, and generate frame-wise captions using GPT-4o. These captions are combined with crowd-sourced task definitions, and then an LLM generates question-answer pairs.
2. QA Refinement: The generated question-answer pairs are filtered by GPT-4o, considering hallucinations and cases where the question does not require visual information in order to be answered. Human validators filter out the invalid question-answer pairs before proceeding to next step.
3. Distractors: LLMs generate distractors (i.e., incorrect answer choices) for the collected question-answer pairs, later, again refined by human validators.

Additionally, the benchmark is divided into 7 sub-categories, which are OCR, Attribute Perception, Object Reasoning, Count Problem, Relationship Recognition, Action Reasoning, and Instructed Description. The benchmarked models are tested under different conditions, which are denoted as customized contexts. The set of customized contexts within this work include multiple videos inputs, long videos, multimodal interleaved input and lastly template video with ImageNet mean pixel values. This work introduces 3 different metrics to measure the positional bias, where all 3 metrics are centered around the relative score (RS) measure. These 3 metrics are, average relative score ( $P_{mean}$ ), difference between maximum and minimum relative score ( $P_{ran}$ ), and the variance in relative scores ( $P_{var}$ ). The evaluations on the proposed benchmark include 27 LVLMs including both open-weight and proprietary models where the model scale is ranging up to 108 billion parameters. Further studies investigates the effect of context type, context length and model size on positional bias.

**Strengths:**

- A novel benchmark for assessing the positional bias in video-language models.
- Focuses an important aspect which could help to assess robustness of LVLMs under different positional settings, which could help further developing more trusthworthy LVLMs.
- Detailed experimentation: 27 models, further beneficial studies on the effect of different design choices.

**Weaknesses:**

- The presentation needs to be improved,
	- In Figure 3, the entire set of sub-tasks should be illustrated. This is not feasible.
	- In Figure 3, the tasks should be renamed by following the terminology already exist in the literature. I don't understand why someone should pose these tasks as *reasoning* tasks, where they are *recognition* tasks in fact. For instance, please see [Fig 1](https://arxiv.org/pdf/2306.13394) in MMU work to see the difference between reasoning and recognition type of tasks. So, the tasks should be renamed as,
		- Object Reasoning -> Object Recognition
		- Count Problem -> Object Counting (because actions can be counted also as well)
		- Action Reasoning -> Action Recognition
		- Attribute Perception -> Attribute Recognition (for consistency)
	- In Fig1(a), what do the numbers next to the tick and x marks?
	- Fig 2 is cluttered, I think it would be better if the only metric is positional variance.
- Human refinement process remains entirely opaque. There are no details on this process, even how many human validators participated in the refinement process.
- The proposed metric is not a standard metric used to evaluate the models, so I think this part should be expanded in the main text. This is such an important element of this paper but it only takes up 20 lines in the main text currently.
	- For instance, why should one use the relative score metric, and why should the standalone accuracy should be in the denominator?
	- What is the position $i$ ? What is the unit, seconds, or frame? Is this position absolute or relative? Are these positions randomly sampled for each individual example? Do these positions guarantee that the question-answer pairs remain valid for the video with custom context?
	- The morphological recognition (MR) term is confusing because morphology is a term which exists in NLP literature. Additionally, MR term currently seems unclear in the main text, and it is not motivated well enough.

**Questions:**

- I think it would be good to show visual examples of customized contexts in the appendix part.
- There is some typo on Fig2: GTP-4o-latest should be GPT-4o-latest.
- There is also white text on the last figure (Fig. 22) which can be barely seen on the background: `"Please output the questions and reference answers in the following JSON format: [ {'question': 'xxx', 'answer': 'xxx’}, {'question': 'xxx', 'answer': 'xxx’},"`

---

> ### Author Response · Authors · 2025-11-25
> **Response to Reviewer TpSR (1/2)**
>
> Dear reviewer TpSR, thank you for your insightful comments. We address your comments point-by-point as follows and have incorporated the corresponding revisions into the manuscript (changes highlighted in blue).
>
> ---
> **Q1:** In Figure 3, the entire set of sub-tasks should be illustrated. This is not feasible.
>
> **A1:** In the initial draft, due to space constraints, Fig. 3 presented only four sub-tasks, while additional sub-tasks were included in the appendix (Fig. 15 and Fig. 16). Thank you for this suggestion. The revised Fig. 3 now includes the full set of sub-tasks.
>
> ---
> **Q2:** In Figure 3, the tasks should be renamed by following the terminology already exist in the literature. I don't understand why someone should pose these tasks as reasoning tasks, where they are recognition tasks in fact. For instance, please see Fig 1 in MMU work to see the difference between reasoning and recognition type of tasks. So, the tasks should be renamed as, Object Reasoning -> Object Recognition; Count Problem -> Object Counting (because actions can be counted also as well); Action Reasoning -> Action Recognition; Attribute Perception -> Attribute Recognition (for consistency)
>
> **A2:** Our original naming was motivated by the following consideration: in our benchmark, each question is formulated using a scene description and a task instruction. LVLMs are required to infer the user-interested segment from diverse contexts based on the scene description; therefore, we adopted the term *reasoning* in the initial draft.
>
> As suggested, since such naming may lead to misunderstanding, we have revised it according to the reviewer’s feedback. Please refer to the revised Fig. 3.
>
> ---
> **Q3:** In Fig1(a), what do the numbers next to the tick and x marks?
>
> **A3:** The numbers in Fig. 1(a) represent the correctness levels of the model’s responses to the question. They serve to intuitively illustrate the uneven comprehension across different contextual positions.
> Thank you for pointing out this potential source of confusion. We have added an explanation to the caption of Fig. 1 in the revised manuscript.
>
> ---
> **Q4:** Fig 2 is cluttered, I think it would be better if the only metric is positional variance.
>
> **A4:** The color scheme used in Fig. 2 of the initial draft may cause visual clutter, which we have addressed in the revision. Reporting composite metrics for evaluation is beneficial, as each metric reflects distinct statistical characteristics. For example, $P_{mean}$ captures the magnitude of positional bias, while $P_{ran}$ and $P_{var}$ quantify the volatility of model behavior. As Reviewer sdzE noted, these metrics collectively offer a holistic assessment.
>
> ---
> **Q5:** Human refinement process remains entirely opaque. There are no details on this process, even how many human validators participated in the refinement process.
>
> **A5:** The construction of Video-LevelGauge involves four major stages requiring human participation. We describe these stages in detail below and have incorporated the corresponding content into the revised manuscript (see Sec. A.8).
>
> - *Video Collection.*
> Diverse videos are manually collected from existing datasets, and blurry, static, and duplicate videos are filtered out. Two annotators participate in this stage, contributing approximately 40 hours of human effort.
>
> - *QA Generation.*
> Two human annotators cross-validate the annotations, generating the task definition along with three positive and three negative QA examples for each video task.
>
> - *QA Refinement.*
> Twelve human annotators further review the QAs that pass the model-based blind filtering and hallucination filtering. Annotators are asked to remove or refine QAs that involve answer leakage, commonsense questions, or answers that are inconsistent with the video content. The twelve annotators are divided into two groups. Each group reviews a portion of the data first, and they then cross-check each other’s assessments. Over 90 hours of human effort are expended in this stage.
>
> - *Distractor Construction.*
> Eight annotators examine the LLM-generated multiple-choice questions, filtering out distractors that are trivially distinguishable or are valid given the video content. MCQAs that do not meet quality requirements are LLM-regenerated until they are rejected after three attempts. The eight annotators are also divided into two groups for cross-validation, contributing approximately 60 hours of human effort.
>
> Human annotators are encouraged to focus on validation and refinement rather than rewriting, thereby reducing their workload.

---

> ### Author Response · Authors · 2025-11-25
> **Response to Reviewer TpSR (2/2)**
>
> **Q6:** The proposed metric is not a standard metric used to evaluate the models, so I think this part should be expanded in the main text. This is such an important element of this paper but it only takes up 20 lines in the main text currently.
>
> **A6:** Thank you for your helpful suggestion. We have expanded this section and added more detailed explanations of each metric and bias pattern in the revised manuscript. Please refer to Sec. 3.4.
>
> ---
> **Q7:** For instance, why should one use the relative score metric, and why should the standalone accuracy should be in the denominator?
>
> **A7:** Using the relative score metric allows us to focus on the effects of varying contextual positions while excluding the influence of the model’s absolute capability and the intrinsic difficulty of different tasks.
>
> Directly comparing raw accuracies would be confounded by inherent differences in model capability. For example, between two models with different standalone performance levels, the lower-performing model may exhibit smaller absolute score differences across contextual positions simply due to its low standalone accuracy.
> Standalone accuracy refers to a model’s performance under interference-free input and reflects its actual capability on our QAs. Using it as the denominator normalizes both question difficulty and model competence, enabling a focused analysis of positional bias. Furthermore, this normalization isolates differences in intrinsic task difficulty, as the sub-tasks are constructed from non-identical video clips.
>
> ---
> **Q8:** What is the position $i$? What is the unit, seconds, or frame? Is this position absolute or relative? Are these positions randomly sampled for each individual example? Do these positions guarantee that the question-answer pairs remain valid for the video with custom context?
>
> **A8:** In Sec. 3.4, without loss of generality, position $i$ denotes the $i$-th position at which the evaluation is performed. In the practical evaluation, as described in Sec. 4.1 (Evaluation Protocol),  we assess ten uniformly distributed positions across the video context. Specifically, $i$ = 1, 2, …, 10, where $i$ = 1 indicates probe insertion at the very beginning of the background video, and $i$ = 10 indicates insertion at the very end. The remaining positions are evenly distributed throughout the video context, with a step size equal to $1/9$ of the total context length.
>
> ---
> **Q9:** The morphological recognition (MR) term is confusing because morphology is a term which exists in NLP literature. Additionally, MR term currently seems unclear in the main text, and it is not motivated well enough.
>
> **A9:** We employ morphological recognition (MR) primarily to delineate different bias patterns.
> As shown in Fig. 7, the model’s positional bias may exhibit various patterns, such as "head preference" and "lost in the middle". To facilitate the identification of such behaviors, rather than relying solely on visual inspection, we introduce MR.
> Together with statistical metrics, MR provides a grounded framework for diagnosing and characterizing positional biases in LVLMs.
>
> As suggested, in view of potential confusion with NLP terminology, we replace MR with Bias Pattern Recognition (BPR) in the revised manuscript.
>
> ---
> **Q10:** I think it would be good to show visual examples of customized contexts in the appendix part.
>
> **A10:** Thank you for your constructive suggestion, we have added visual examples of the six customized contexts in the appendix. Please refer to Fig. 14.
>
> ---
> **Q11:** There is some typo on Fig2: GTP-4o-latest should be GPT-4o-latest. There is also white text on the last figure (Fig. 22) which can be barely seen on the background: "Please output the questions and reference answers in the following JSON format: [ {'question': 'xxx', 'answer': 'xxx’}, {'question': 'xxx', 'answer': 'xxx’},"
>
> **A11:** We sincerely appreciate your careful review. These typos have been corrected in the revision.
>
> ---
> We deeply appreciate the reviewer's time and thoughtful suggestions. We look forward to addressing any additional questions or providing further clarification needed.

---

> > ### Comment · Reviewer_TpSR · 2025-11-26
> >
> > I thank the authors for their response. I decided to increase my rating to 4 as most of my concerns were addressed by the authors. I have another question about the metric. I understand that there should be _relative_ score but I don't understand why it should be $\frac{S_i}{S_{meta}}$ and not $\left|S_{i} - S_{meta}\right|$. Can you elaborate more on this?
> >
> > Minor suggestions for a better presentation: As the current version still below 10 full pages, I'd increase the font size in Fig. 3. Actually, it'd be better split it into 2 figures: Keep (a) and (b) in one figure with full width and (c) becomes a non-full width figure like Fig. 5 in the current version. It is also difficult read the text and numbers in Fig. 6 and Fig. 7: one could decrease the number of ticks in y-axis for instance and increase the font size.

---

> > > ### Author Response · Authors · 2025-11-27
> > > **Response to Reviewer TpSR**
> > >
> > > Dear reviewer TpSR, thank you for the raised score! We address your new questions below:
> > >
> > > ---
> > >
> > > **Q1:** I understand that there should be relative score but I don't understand why the *relative score* should be $\frac{S_{i}}{S_{meta}}$ and not $|S_{i}-S_{meta}|$.
> > >
> > > **A1:** $\frac{S_{i}}{S_{meta}}$ normalizes the score by the model’s intrinsic capability, whereas $|S_{i}-S_{meta}|$, despite subtracting $S_{meta}$, still reflects an absolute score.
> > >
> > > To illustrate this distinction, we provide examples of two models below. Although these may be somewhat extreme cases, they make the intuition more transparent.
> > >
> > > | Model   | $S_{meta}$ | $S_{i}$ | $\|S_{i}-S_{meta}\|$ |
> > > |:---------:|:------------:|:------------:|:------------:|
> > > | Model A | 0.9        | 0.8     | 0.1                |
> > > | Model B | 0.4        | 0.3     | 0.1                |
> > >
> > > Although the absolute differences (0.1) appear identical for the two models, in reality Model B experiences a $1/4$ performance drop, whereas Model A drops by only about $1/9$. The absolute difference inadequately supports cross-model comparisons (nor comparisons across subtasks). *Relative score* represents how much the performance deviates relative to the model’s intrinsic capability (i.e., proportionally).
> > >
> > > Moreover, normalizing with $S_{meta}$ maps the *relative scores* of different models (or subtasks) into a common range from 0 to 1, enabling more convenient and elegant comparisons, such as in scatter-plot visualizations.
> > >
> > > ---
> > >
> > > **Q2:** Minor suggestions for a better presentation: As the current version still below 10 full pages, I'd increase the font size in Fig. 3. Actually, it'd be better split it into 2 figures: Keep (a) and (b) in one figure with full width and (c) becomes a non-full width figure like Fig. 5 in the current version. It is also difficult read the text and numbers in Fig. 6 and Fig. 7: one could decrease the number of ticks in y-axis for instance and increase the font size.
> > >
> > > **A2:** We sincerely appreciate your constructive comments that help improve the quality of our manuscript. The improvements you suggested have been fully implemented. Please see the updated version.
> > >
> > > ---
> > > Thanks again for your valuable feedback. Feel free to let us know if you have any further questions. We are looking forward to your feedback!

---

### Official Review · Reviewer_ZbYf · 2025-10-28

**Soundness:** 3
**Presentation:** 3
**Contribution:** 3
**Rating:** 4
**Confidence:** 4

**Summary:**

This paper introduces Video-LevelGauge, a benchmark exploring positional bias in large video language models (LVLMs). The benchmark covers a range of models for comprehensive evaluation and reveals that existing open-sourced ones suffer significant positional bias, exhibiting unstabe video understanding affected by the position of target content. The paper also provides a deep analysis on models’ behavior across divers perspectives, such as context length and model scales.

**Strengths:**

- The paper tackles an important and underexplored problem in video understanding — positional bias in LVLMs. Overall, the paper is well written and easy to follow.
- The benchmark is well structured, featuring diverse videos and mechanisms that prevent single-frame or blind-bias predictions. The positional bias metric is clearly defined and suitable for evaluation.
- The paper provides extensive experimental results across 27 leading LVLMs with in-depth analyses, offering valuable insights into model behavior.

**Weaknesses:**

- While the benchmark design and evaluation criteria are solid, the results and analyses feel somewhat lukewarm and unsurprising. In Section 4.2, the main conclusion that positional bias tendencies vary across models and depend largely on training methods and model scale is expected. It is natural that larger models exposed to diverse video data perform better, and this conclusion could likely apply to other evaluation criteria beyond positional bias. I would appreciate a deeper, more tailored analysis of *why* such biases arise and *what* model characteristics contribute to distinct bias patterns (e.g., MR types). For instance, why do MiniGPT4-Video and InternVL3 (8B) show head preference? Are their training datasets skewed toward early-frame cues?
- Consequently, it seems somewhat trivial that longer videos are more challenging and lead to lower performance. However, it remains unclear whether this truly amplifies positional bias or if the observed drop is simply due to increased video complexity.
- The paper primarily identifies the problem of positional bias but does not propose or discuss concrete directions for mitigation. Including such discussion would make the contribution more complete and valuable for future work.
- (Minor) Discussing inconsistent and biased video understanding found in prior works [1, 2] could also strengthen this analysis.

**References**

[1] A Closer Look at Temporal Sentence Grounding in Videos: Dataset and Metric, arxiv 2021

[2] On the Consistency of Video Large Language Models in Temporal Comprehension, CVPR 2025

**Questions:**

See the Weaknesses

---

> ### Author Response · Authors · 2025-11-25
> **Response to Reviewer ZbYf  (1/2)**
>
> Dear reviewer ZbYf, thank you for your insightful comments. We address your comments point-by-point as follows and have incorporated the corresponding revisions into the manuscript (changes highlighted in blue).
>
> ---
> **Q1:** While the benchmark design and evaluation criteria are solid, the results and analyses feel somewhat lukewarm and unsurprising. In Section 4.2, the main conclusion that positional bias tendencies vary across models and depend largely on training methods and model scale is expected. It is natural that larger models exposed to diverse video data perform better, and this conclusion could likely apply to other evaluation criteria beyond positional bias. I would appreciate a deeper, more **tailored analysis** of why such biases arise and **what model characteristics contribute to distinct bias patterns** (e.g., MR types). For instance, why do MiniGPT4-Video and InternVL3 (8B) show head preference? Are their training datasets skewed toward early-frame cues?
>
> **A1:** Thank you for this helpful question. Beyond the general analysis of existing models in Sec. 4.2, we provide more tailored analysis of the key factors influencing positional bias in Sec. 4.3, which covers context categories, multi-video scenarios, and context length. In addition, Sec. A.1 and Sec. A.2 further present comparisons with LLM positional biases and an analysis of hour-long video settings, respectively.
>
> Experiments show that positional bias in LVLMs stems from multiple factors:
>
> - **Training context length.**
> As shown in Fig. 7, positional-bias patterns shift under different context-length settings. When the inference context exceeds the model’s training-context window, positional bias becomes more severe, indicating that the training-context length contributes to the bias. Image-oriented models (e.g., *Qwen-VL-Max*) show pronounced positional bias, whereas models trained with extensive long-video data (e.g., *Video-XL2*) demonstrate much lower bias.
>
> - **Model optimization and inference mode.** *VideoRefer*, optimized for fine-grained spatial-temporal understanding, exhibits markedly lower positional bias than models of similar size. Furthermore, for *MiMo-VL* and *GLM-4.5V*, the reasoning mode mitigates positional bias relative to non-reasoning modes (Fig. 13).
>
> - **Annotation bias in training data.** Annotation bias in training corpora is a significant contributor. For instance, *MiniGPT4-Video* is trained on *WebVid*, which suffers from static appearance bias [1], where early frames alone often suffice for understanding.
>
> - **Interleaved multimodal inputs.** Our experiments show that video-text interleaved inputs lead to severe positional bias, likely due to insufficient interleaved training data, revealing model brittleness in complex contextual scenarios.
>
> - **Inherited LLM biases.** Since most LVLMs are built on top of an LLM component, they inherit language model positional biases to some extent. A detailed comparative analysis is provided in Sec. A.1.
>
> - **Task differences.** LVLMs display varying levels of positional bias across tasks (see Fig. 12), further underscoring that positional bias is shaped by both model characteristics and task-specific capabilities.
>
> As suggested, we have refined the analysis in Sec. 4.2 accordingly. Please refer to the revised manuscript.
>
> ---
> **Q2:** Consequently, it seems somewhat trivial that longer videos are more challenging and lead to lower performance. However, it remains unclear whether this truly amplifies positional bias or if the observed drop is simply due to increased video complexity.
>
> **A2:** We conducted comprehensive investigations of long-video scenarios and isolated the underlying contributing factors. Our experiments demonstrate that the intensified positional bias in long-video settings is primarily driven by two factors. (1) Long videos (with more sampling frames) naturally produce longer contexts, which may exceed the model’s typical training context window, thus amplifying positional bias (Fig. 7). (2) Long videos typically contain more complex and diverse visual contents, and such contextual complexity substantially contributes to positional bias (Fig. 6).
>
> More specifically, as shown in Fig. 6, when controlling for context length by uniformly sampling the same number of frames, LVLMs exhibit similar levels of positional bias in multi-video and long-video scenarios, levels considerably higher than those observed in short-video settings. This indicates that contextual diversity is a key contributor to positional bias. Furthermore, for the same long videos, we experiment with an fps-prioritized sampling scheme, which further increases context length and consequently intensifies positional bias (Fig. 9), consistent with the context-length analysis in Fig. 7.
>
> In conclusion, positional bias intensifies with increasing context length and contextual complexity, both inherent characteristics of long videos.

---

> ### Author Response · Authors · 2025-11-25
> **Response to Reviewer ZbYf (2/2)**
>
> **Q3:** The paper primarily identifies the problem of positional bias but does not propose or discuss concrete directions for mitigation. Including such discussion would make the contribution more complete and valuable for future work.
>
> **A3:** Thank you for your constructive suggestions. Building on our experimental findings, we discuss potential directions for mitigating positional bias in Sec. 4.4 of the manuscript.
>
> - For training, we observe that positional bias increases in long-video scenarios and in video-text interleaved contexts. Post-training on long-video or interleaved video–text data demonstrates promising potential for mitigation.
>
> - For inference optimization, positional bias intensifies as context length exceeds the typical training context length,, suggesting that effective training-free video token ensemble strategies and slow-fast two-stream features hold promise for mitigation.
>
> - Model Architecture optimization, such as improved multimodal positional encoding and question-guided context attention mechanisms, represents a potential avenue for improvement.
>
> As the first comprehensive study on positional bias in video understanding, this work focuses primarily on constructing a dedicated benchmark and metric, thereby providing a framework to diagnose and characterize positional biases. We conducted extensive evaluations of existing models, investigated multiple contributing factors, analyzed positional-bias behaviors across different real-world scenarios, and further compared it to positional bias in language models (Sec. A.1).
>
> ---
> **Q4:** (Minor) Discussing inconsistent and biased video understanding found in prior works [2, 3] could also strengthen this analysis.
>
> **A4:** Thank you for pointing out the relevant works. Work [2] investigates moment annotation biases in video grounding, and work [3] examines prediction inconsistencies of LVLMs in video temporal grounding tasks. These studies offer valuable insights. Particularly, the shifted grounding evaluation in work [3] reveals that LVLMs may rely on the linguistic priors of the query for grounding rather than genuine visual understanding. This indicates the presence of positional preferences in video grounding tasks, likely stemming from annotation biases in the training data. These findings and our work collectively highlight the value of conducting a thorough and nuanced evaluation of LVLMs. We have incorporated a discussion of these works into the revised manuscript. Please refer to Sec. 4.4.
>
> ---
> **References**
>
> [1] Jie Lei, Tamara Berg, and Mohit Bansal. 2023. Revealing Single Frame Bias for Video-and-Language Learning. In Proceedings of the 61st Annual Meeting of the Association for Computational Linguistics (Volume 1: Long Papers), pages 487–507, Toronto, Canada. Association for Computational Linguistics.
>
> [2] Yitian Yuan, Xiaohan Lan, Xin Wang, Long Chen, Zhi Wang, and Wenwu Zhu. A closer look at tem-
> poral sentence grounding in videos: Dataset and metric. In Proceedings of the 2nd international
> workshop on human-centric multimedia analysis, pp. 13–21, 2021.
>
> [3] Minjoon Jung, Junbin Xiao, Byoung-Tak Zhang, and Angela Yao. On the consistency of video large
> language models in temporal comprehension. In Proceedings of the Computer Vision and Pattern
> Recognition Conference, pp. 13713–13722, 2025.
>
> ---
> We sincerely thank the reviewer for their valuable time and constructive feedback. We remain open to addressing any further questions or concerns the reviewer may have regarding our revised manuscript.

---

> ### Comment · Reviewer_ZbYf · 2025-11-26
> **Official Comments by Reviewer ZbYf**
>
> I appreciate the authors' detailed responses to my earlier concerns, and I also acknowledge that several factors may exacerbate and contribute to the positional bias problem.
>
> Most concerns are addressed, but still further clarifications are much needed:
> - While Video-LevelGauge covers diverse domains, probes, and context types relevant to the proposed problem, its overall size (438 videos) remains relatively small compared with existing benchmarks. For instance, Video-MME includes 900 videos with human-annotated QA pairs. I acknowledge that Video-LevelGauge follows different design criteria than modern benchmarks, but I would suggest that the authors clarify and emphasize how firmly the current dataset supports the conclusions drawn. This would frame the novelty better.
> - Secondly, as other reviewers have noted, I remain unconvinced that the relative score provides a fully fair basis for model comparison. The metric in Table 1 appears to depend on the absolute number of $S_{meta}$. In addition, I would like to know whether the authors exclude cases in which a model fails to answer the standalone input. If such cases are included, the analysis may reflect empty patterns rather than genuine video understanding.
> - (Minor) It would be helpful to include text-only methods in Table 1. They can serve as a lower-bound baseline and help readers better interpret the severity of positional bias in current models.

---

> > ### Author Response · Authors · 2025-11-27
> > **Response to Reviewer ZbYf**
> >
> > Dear reviewer ZbYf, thank you for your valuable feedback. We address your further questions below:
> >
> > ---
> > **Q1:** While Video-LevelGauge covers diverse domains, probes, and context types relevant to the proposed problem, its overall size (438 videos) remains relatively small compared with existing benchmarks. For instance, Video-MME includes 900 videos with human-annotated QA pairs. I acknowledge that Video-LevelGauge follows different design criteria than modern benchmarks, but I would suggest that the authors clarify and emphasize how firmly the current dataset supports the conclusions drawn. This would frame the novelty better.
> >
> > **A1:** Video-LevelGauge consists of 438 videos and 1,177 MCQAs built upon them. Compared with Video-MME, which contains 900 videos and 2,700 QAs, its scale is relatively smaller. We would like to clarify that we intentionally keep the number of probes moderate, although constructing more probes does not require substantial annotation labor under the adopted *standardized probe and customized context* paradigm. To evaluate positional bias, each probe needs to be inserted into multiple context positions, thereby expanding the scale of evaluation. For example, with 10 positions by default, evaluating 1,177 QA pairs results in over 10,000 model inferences.
> >
> > This is also why we primarily adopt the MCQA format, as its evaluation is significantly faster than open-end questions. To enhance the usability of the benchmark, we intentionally keep the number of probes moderate, and allocate more effort to improving their quality. For instance, we focus on preventing single-frame or blind-bias predictions, and to suppress potential context leakage (Sec. 3.2.3). These quality-validation efforts are essential for ensuring our reliable assessment of positional bias.
> >
> > Finally, the probes are highly extensible. They serve as “seeds” that can be flexibly inserted at arbitrary positions, with adjustable context lengths and categories depending on the testing scenario. The key intuition is that the model should be able to comprehend any part of the context to answer the questions, given that the relevant content may appear anywhere in the sequence.
> >
> > ---
> > **Q2:** Secondly, as other reviewers have noted, I remain unconvinced that the relative score provides a fully fair basis for model comparison. The metric in Table 1 appears to depend on the absolute number of $S_{meta}$. In addition, I would like to know whether the authors exclude cases in which a model fails to answer the standalone input. If such cases are included, the analysis may reflect empty patterns rather than genuine video understanding.
> >
> > **A2:** *Relative score* represents how much the performance deviates proportionally relative to the model’s intrinsic capability. By normalizing with the model’s standalone performance, we compare the relative degree of deviation across different models, ensuring fairness by decoupling positional bias from their absolute capability levels. In Table 1, $S_{meta}$ measures a model’s accuracy on the probes and reflects its conventional visual understanding ability, which traditional benchmarks focus on. Although a strong model may be strong in various aspects, $S_{meta}$ is not correlated with positional bias. For example, Qwen-VL-Max, as an image-oriented model, attains a high $S_{meta}$, yet still exhibits severe positional bias.
> >
> > We assume your question refers to excluding samples for which the model fails on the standalone input, and evaluating positional bias only on the subset of samples that the model answers correctly. We do not perform such filtering, and the test datas are kept identical for all models. Model-dependent data filtering can inadvertently favor certain model preferences and is therefore unsuitable for fair evaluation. Positional bias is a statistical property of a model’s accuracy distribution across positions. We ensure independence between data and model behaviors, and compute positional bias using the complete set of samples.
> >
> > ---
> > **Q3:** (Minor) It would be helpful to include text-only methods in Table 1. They can serve as a lower-bound baseline and help readers better interpret the severity of positional bias in current models.
> >
> > **A3:** Thank you for this constructive comment. As suggested, we have added text-only results for each model in Table 1, which serve as lower-bound baselines. Please refer to the updated version. To facilitate comparison with the average positional-bias score $P_{mean}$, we consistently normalize the text-only accuracy using $S_{meta}$, denoted as $P_{text}$. For most models, there is a clear margin between $P_{mean}$ and $P_{text}$, indicating that although positional bias exists, the models do not exhibit complete blindness to the context.
> >
> > ---
> > Thanks again for your valuable feedback. Feel free to let us know if you have any further questions. We are looking forward to your feedback!

---

> > > ### Comment · Reviewer_ZbYf · 2025-11-28
> > > **Official Comments by Reviewer ZbYf**
> > >
> > > I appreciate the authors' effort in addressing my concerns. Overall, I have no major concerns and decided to raise the score.
> > >
> > > Please consider including the above discussions in the final manuscript.

---

### Official Review · Reviewer_3Lzq · 2025-10-29

**Soundness:** 2
**Presentation:** 2
**Contribution:** 2
**Rating:** 6
**Confidence:** 2

**Summary:**

This paper introduces Video-LevelGauge, a benchmark for evaluating contextual positional bias in LVLMs by inserting standardized probes into different positions of video contexts. The benchmark systematically assesses 27 LVLMs, revealing that commercial models exhibit less positional bias than open-source ones. The study also explores the effects of context type, length, and model size on positional bias, providing insights for future improvements.

**Strengths:**

- Novel Contribution: Highlights contextual positional bias, an underexplored issue in LVLMs.
- Well-Designed Benchmark: Includes standardized probes, flexible context configurations, and comprehensive metrics.
- Comprehensive Evaluation: Thorough analysis of 27 LVLMs, revealing actionable insights on bias patterns.

**Weaknesses:**

- No Mitigation Strategies: While the paper introduces contextual positional bias, it does not propose or evaluate methods to mitigate it.
- Lack of Task-Specific Examples: The paper does not provide clear illustrations of how positional bias affects specific tasks, making the findings less interpretable.
- Narrow Scope: Focuses solely on positional bias without addressing other LVLM limitations, such as hallucination or temporal reasoning errors.

**Questions:**

Please see Weaknesses for details.

---

> ### Author Response · Authors · 2025-11-25
> **Response to Reviewer 3Lzq**
>
> Dear reviewer 3Lzq, thank you for your insightful comments. We address your comments point-by-point as follows and have incorporated the corresponding revisions into the manuscript (changes highlighted in blue).
>
> ---
> **Q1:** No Mitigation Strategies: While the paper introduces contextual positional bias, it does not propose or evaluate methods to mitigate it.
>
> **A1:**
> Based on the evaluation of positional bias across 27 different models and comprehensive analyses of its contributing factors, we discuss potential mitigation directions in Sec. 4.4.
>
> - For training, we observe that positional bias worsens in long-video settings and in video–text interleaved contexts. Post-training on long-video or interleaved video–text data appears to be a promising avenue for mitigation.
>
> - For inference optimization, positional bias becomes more pronounced as context length increases, suggesting that effective video-token ensemble strategies and slow–fast two-stream features may help alleviate this issue.
>
> - Model architecture optimization, such as improved multimodal positional encoding and question-guided context-attention mechanisms, can be potential directions for improvement.
>
> As a pioneering study on positional bias in video understanding, we are dedicated to construct a comprehensive benchmark, enhance customizability, validate probe requirements, isolate contextual
> interference, and establish a standardized positional bias metric, thereby providing a framework for diagnosing and characterizing positional biases. We conduct extensive evaluations of existing models and actively investigate how such biases manifest across diverse real-world scenarios, offering actionable insights for future model optimization.
>
> ---
> **Q2:** Lack of Task-Specific Examples: The paper does not provide clear illustrations of how positional bias affects specific tasks, making the findings less interpretable.
>
> **A2:** In Fig. 12 of the appendix, we provide a visualization of positional bias across six video sub-tasks. It demonstrates the strong performance of Qwen2.5-VL-7B on the OCR task, where the model exhibits minimal positional bias. In contrast, it reveals pronounced positional bias in relationship recognition and attribute recognition tasks. Qwen2.5-VL-7B shows consistent and effective OCR capability across the entire sequence and uneven capability on relationship and attribute recognition, i.e., exceling at visual content appearing at the beginning and end of the video compared to the middle section.
>
> Notably, relative scores are employed, meaning that the model’s scores for each task at various positions are normalized (Eq. 1). This normalization isolates differences in intrinsic task difficulty, as the sub-tasks are constructed from non-identical video clips. The reported relative scores therefore reflect solely the influence of contextual position.
>
> ---
> **Q3:** Narrow Scope: Focuses solely on positional bias without addressing other LVLM limitations, such as hallucination or temporal reasoning errors.
>
> **A3:**
> Thank you for your advice. Evaluating these limitations is indeed valuable for improving LVLMs. Our Video-LevelGauge is specifically designed to evaluate positional bias in video understanding and multi-modal interleaved scenarios.
>
> Within Video-LevelGauge, we (1) construct seven subtask-oriented QAs annotated on diverse videos; (2) validate the quality of the constructed QAs and examine potential contextual leakage; (3) propose four types of customized contexts to simulate different real-world scenarios; and (4) introduce rigorous metrics and analytical methods for positional-bias assessment.
> In this way, we evaluate 27 existing models and conduct an in-depth analysis of the factors influencing positional bias, including task type, context length, model size, and multimodal reasoning. We further investigate how positional bias manifests across a variety of realistic settings, such as long-video scenarios, short-video scenarios, multi-video contexts, and video–text interleaving inputs.
>
> Existing benchmarks, such as VidHalluc [1] and TempCompass [2], are dedicated to evaluate hallucination and temporal reasoning errors, and they hold irreplaceable value. As you mentioned in the *Strengths*, "positional bias in video understanding remains underexplored issue."  Video-LevelGauge is fully devoted to investigating this issue and thus serves as a complement to existing efforts.
>
> [1] Li C, Im E W, Fazli P. Vidhalluc: Evaluating temporal hallucinations in multimodal large language models for video understanding. Proceedings of the Computer Vision and Pattern Recognition Conference. 2025: 13723-13733.
>
> [2] Liu Y, Li S, Liu Y, et al. TempCompass: Do video LLMs really understand videos? Findings of the Association for Computational Linguistics: ACL 2024, pp. 8731–8772
>
> ---
> We sincerely thank the reviewer for their time and constructive comments. We would like to address any further questions or concerns the reviewer may have.

---

> > ### Comment · Reviewer_3Lzq · 2025-11-26
> > **RE: Rebuttal**
> >
> > Thank you for the response, which has addressed my concerns. I will maintain my original positive rating with higher confidence. I also hope that if time allows, the authors might consider exploring preliminary mitigation strategies in a future version.

---

### Author Response · Authors · 2025-12-01
**Paper summary and major improvements for Area Chair**

Dear AC,

We summarize our paper content and major improvements during the discussion period as below.

**Paper summary:**\
This paper focuses on an underexplored issue in video understanding - contextual positional bias of Large Video Language Models (LVLMs). LVLMs commonly exhibit uneven comprehension of visual content presented at different contextual positions, a problem less revealed by existing benchmarks.

We introduce Video-LevelGauge, a dedicated benchmark designed to systematically assess positional bias across diverse tasks, video types, and real-world scenarios. In addition, we introduce a comprehensive analysis method that combines statistical measures with bias pattern recognition, providing a grounded framework to diagnose and characterize positional biases in LVLMs.
We systematically assess 27 LVLMs, and also provide a deep analysis on models’ behavior across diverse perspectives, such as context types, context length, and model scales, offering valuable insights into model behavior.

Finally, based on the extensive evaluation, we discuss potential mitigation directions. Given that the model should be able to comprehend any part of the context to answer the questions, since the relevant content may appear anywhere in the sequence, we believe Video-LevelGauge can serve as a useful complement to existing benchmarks.


**Major improvements:**\
During the discussion period, we received many valuable comments from the reviewers, which significantly helped us improve the quality of our manuscript. We sincerely appreciate their constructive suggestions. The revisions are highlighted in blue in the rebuttal revision, and we summarize them as follows:

- *Improvements in presentation.*
(1) We provided a more detailed explanation of our evaluation metrics, including the design motivation and implementation (Sec. 3.4).
(2) We revised several task names and metric terms to avoid potential confusion, such as renaming morphological recognition (MR) to Bias Pattern Recognition (BPR).
(3) We split the original Figure 3 into two figures to more clearly present the subtask examples.

- *Improvements in completeness.*
(1) We added additional details about the human refinement process (Sec. A.8).
(2) We clarified the design criteria of the benchmark and emphasized how firmly our compact benchmark supports the evaluation of positional bias (Sec. A.9).
(3) We provided validation of potential leakage between the probe clip and background videos (Table 5).
(4) We included visual examples of the six types of customized contexts (Figure 15).

- *More comprehensive analyses.*
(1) We added the text-only performance of each model to serve as a lower-bound baseline for positional bias analysis (Table 1).
(2) We clarified the independence between $S_{meta}$ and positional bias, noting that models with high $S_{meta}$ can still exhibit severe positional bias.
(3) Drawing on literature from video grounding, we analyzed potential origins of positional bias from the perspective of annotation biases.

We hope that this summary can help you gain a clearer understanding of our work. Thank you very much for your valuable time and consideration.

Sincerely, \
Authors of submission 1486.

---

### Author Response · Authors · 2025-12-02
**Summary comment for Area Chair**

Dear AC,

During the discussion period, all four reviewers (initially two positive and two negative) indicated that our responses had addressed their major concerns.

Positive reviewer #3Lzq maintained their positive rating and raised their confidence after exchanging with us, and the other positive reviewer #sdzE kept their rating.
Notably, both negative reviewers (#ZbYf and #TpSR) raised their ratings during the discussion.
In particular, we addressed most concerns of reviewer #TpSR, such as the motivation behind our metrics. Their impression of the paper improved markedly. The remaining minor formatting concerns have also been fully resolved. There is a potential for them to further improve their rating.

Unfortunately, the discussion period was cut short.
We sincerely and respectfully invite you to take a moment to review our scientifically grounded discussions.
Thank you very much for your valuable time and consideration.

Sincerely, \
Authors of submission 1486.

---

### Meta-Review · Area_Chair_sdDR · 2026-01-12

**Summary:**

The initial reviews presented a split decision with two positive and two negative scores. The core concerns that informed the initial suggested decision (likely leaning towards rejection or weak acceptance) were:

- The lack of proposed mitigation strategies for the identified bias, unclear task-specific examples, and a narrow focus solely on positional bias, without addressing other limitations of LVLM.

- A deeper, tailored analysis of the origins of distinct bias patterns is requested, and questioned whether longer videos truly amplify positional bias or just lower overall performance. Noted the absence of concrete mitigation directions.

- Concerns about presentation clarity (figure design, task naming, opaque human refinement process), the motivation and explanation of the proposed non-standard metrics, and terminology confusion.

- A lack of details on metric definitions (especially edge cases) and clarification of bias pattern categories. Also, some technical concerns about potential information leakage between probe clips and background videos, which could invalidate results.

**Reviewer Concerns:**

Addressed Concerns:

- Reviewer 3Lzq: Most major concerns were addressed.

- Reviewer ZbYf: This reviewer acknowledged the detailed response and raised their score.

- Reviewer TpSR: The authors comprehensively addressed almost all presentation issues. The reviewer was satisfied enough to raise their score.

- Reviewer sdzE: The reviewer stated they had no major concerns.

Unaddressed Concerns:

- Reviewer ZbYf: Even after raising their score, the reviewer had residual concerns.

- Mitigation Strategies: Reviewers 3Lzq and ZbYf both highlighted the lack of concrete mitigation methods as a weakness. The authors' position is that proposing such methods is beyond the scope of this diagnostic benchmark paper.

**Reviewer Scores:**

Reviewer ZbYf and Reviewer TpSR may raise their scores, while others will keep the scores.

---

### Decision · Program_Chairs · 2026-01-26

Accept (Poster)